# OCTAX: ACCELERATED CHIP-8 ARCADE ENVIRONMENTS FOR REINFORCEMENT LEARNING IN JAX

**Waris Radji**[1]*    **Thomas Michel**[1]    **Hector Piteau**[2]

[1]Univ. Lille, Inria, CNRS, Centrale Lille, UMR 9189-CRIStAL, France
[2]Independent Researcher

{waris.radji,thomas.michel}@inria.fr   hector.piteau@gmail.com

## ABSTRACT

Reinforcement learning (RL) research requires diverse, challenging environments that are both tractable and scalable. While modern video games may offer rich dynamics, they are computationally expensive and poorly suited for large-scale experimentation due to their CPU-bound execution. We introduce OCTAX, a high-performance suite of classic arcade game environments implemented in JAX, based on CHIP-8 emulation, a predecessor to Atari, which is widely adopted as a benchmark in RL research. OCTAX provides the JAX community with a long-awaited end-to-end GPU alternative to Atari games, offering image-based environments, spanning puzzle, action, and strategy genres, all executable at massive scale on modern GPUs. Our JAX-based implementation achieves orders-of-magnitude speedups over traditional CPU emulators. We demonstrate OCTAX's capabilities by training RL agents across multiple games, showing significant improvements in training speed and scalability compared to existing solutions. The environment's modular design enables researchers to easily extend the suite with new games or generate novel environments using large language models, making it an ideal platform for large-scale RL experimentation. Our open-source framework is available at https://github.com/riiswa/octax/.

## 1 INTRODUCTION

Modern reinforcement learning (RL) research (Sutton & Barto, 2018) demands extensive experimentation to achieve statistical validity, yet computational constraints severely limit experimental scale. RL papers routinely report results with fewer than five random seeds due to prohibitive training costs (Henderson et al., 2018; Colas et al., 2018; Agarwal et al., 2021; Mathieu et al., 2023; Gardner et al., 2025). While understandable from a practical standpoint, this undersampling undermines statistical reliability and impedes algorithmic progress. Environment execution creates this bottleneck: while deep learning has embraced end-to-end GPU acceleration, RL environments remain predominantly CPU-bound. Originally designed under severe hardware constraints, classic arcade games represent a solution for scalable RL experimentation. The Atari Learning Environment (ALE) (Bellemare et al., 2013) has established itself as a standard RL benchmark, although existing implementations remain fundamentally CPU-bound. As noted by Obando-Ceron & Castro (2020), the Rainbow paper (Hessel et al., 2018) required 34,200 GPU hours (equivalent to 1,425 days) of experiments, a computational cost that is prohibitively high for small research laboratories. In this paper, we propose an alternative approach for training RL agents in environments that share mechanisms with ALE, but which is not intended as a drop-in replacement and offers significantly reduced computational cost.

**Contributions.**    We introduce OCTAX, a suite of arcade game environments implemented in JAX (Bradbury et al., 2018a) through CHIP-8 emulation. CHIP-8, a 1970s virtual machine specification contemporary with early Atari systems, became the foundation for numerous classic games spanning puzzle, action, and strategy genres. CHIP-8's constraint-driven design creates games with similar

---

*The project page is available at https://warisradji.com/octax-page/

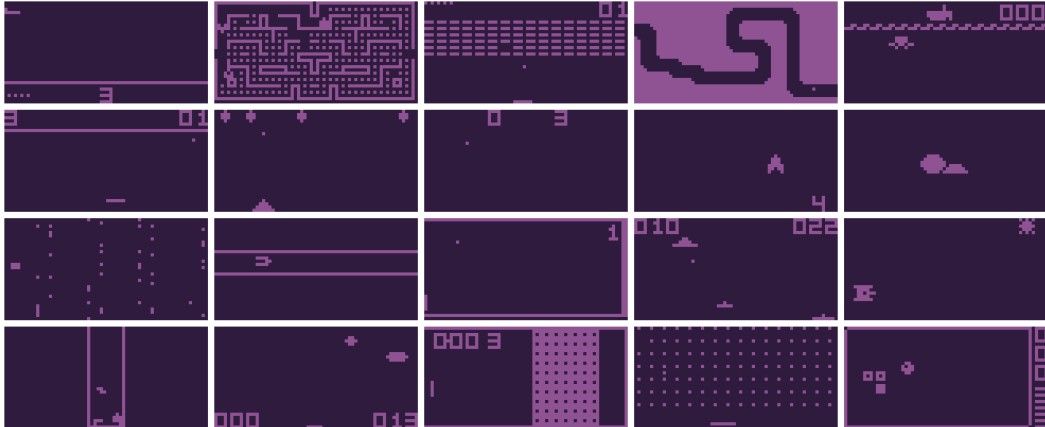

Figure 1: Overview of CHIP-8 game environments implemented in OCTAX.

cognitive demands to Atari while enabling efficient vectorized emulation that scales to thousands of parallel instances. The JAX ecosystem has rapidly emerged as a solution for scalability in RL research but lacks native environments, particularly image-based ones. Our framework addresses this gap by transforming classic games into fully vectorized, GPU-accelerated simulations. These simulations run thousands of game instances in parallel while maintaining perfect fidelity to the original mechanics. This approach dramatically reduces experiment times. Experiments that previously required days or weeks can now be completed in hours. This efficiency makes comprehensive hyperparameter sweeps and ablation studies computationally feasible. The modular design facilitates extension with new games or automated generation using large language models that can directly output CHIP-8 assembly code. Figure 1 provides an overview of the integrated CHIP-8 games.

**Outline.** First, we present our end-to-end JAX implementation of classic arcade environments through CHIP-8 emulation (Section 3). Second, we demonstrate diverse learning dynamics through PPO evaluation across 16 games (Section 4.1). Third, we achieve 350,000 environment steps per second (1.4 million frames per second) on consumer-grade hardware, substantially outperforming CPU-based solutions (Section 4.2). Fourth, we establish an LLM-assisted pipeline for automated environment generation that creates meaningful difficulty gradients (Section 4.3).

## 2  RELATED WORK

Game environments have proven essential for RL research because they provide engaging, human-relevant challenges with clear success metrics. The Arcade Learning Environment (ALE) Bellemare et al. (2013) demonstrated this principle by establishing Atari 2600 games as the standard RL benchmark, enabling breakthrough algorithms like DQN (Mnih et al., 2015) and Rainbow (Hessel et al., 2018). The success of these classic arcade games stems from their constraint-driven design: simple rules that yield complex behaviors, deterministic dynamics that enable reproducible experiments, and visual complexity that tests spatial reasoning without overwhelming computational resources. While algorithmic advances demand increasingly large-scale experiments with dozens of parallel environments and extensive hyperparameter sweeps, traditional game environments remain CPU-bound and poorly suited for parallel execution. This mismatch has driven a progression of solutions, each addressing different aspects of the scalability problem.

**Game-based RL environment platforms.** Increasingly sophisticated gaming platforms have been developed to test different dimensions of learning performance. NetHack Learning Environment (Küttler et al., 2020) provides procedurally generated roguelike challenges that test long-term planning, while Crafter (Hafner, 2021) offers simplified Minecraft-like environments focused on resource management. These environments expand cognitive challenges beyond arcade games, but their CPU-based implementations compound the scalability problem.

**CPU high-performance solutions.** Several projects have focused on optimizing CPU-based environment execution. EnvPool (Weng et al., 2022) achieves substantial speed improvements through highly optimized C++ implementation, demonstrating up to 1 million Atari frames per second on high-end hardware. PufferLib (Suarez, 2025) provides environments written entirely in C, achieving millions of steps per second through over 20,000 lines of optimized code. While these approaches improve CPU throughput, they retain fundamental limitations: costly CPU-GPU data transfers during training and require C implementation in a Python-dominated field.

**GPU-accelerated RL environments.** GPU-accelerated solutions target the constraint more directly by moving environment execution to accelerators. CUDA Learning Environment (CuLE) (Dalton et al., 2020) provides a pioneering CUDA port of ALE, achieving 40-190 million frames per hour on single GPUs. Isaac Gym (Makoviychuk et al., 2021) demonstrates similar principles for robotics tasks, achieving 2-3 orders of magnitude speedups over CPU approaches by running thousands of environments simultaneously. These GPU approaches solve computational bottlenecks but introduce NVIDIA hardware dependence and substantial per-environment engineering costs.

**JAX-based environments.** The adoption of JAX (Bradbury et al., 2018b) has enabled natively accelerated environments that combine portability across hardware with end-to-end GPU acceleration. Brax (Freeman et al., 2021) established viability through MuJoCo-like physics simulation, while Gymnax (Lange, 2022) provides JAX implementations of classic control tasks and simplified environments from BSuite (Osband et al., 2019) and MinAtar (Young & Tian, 2019). Specialized environments target specific research needs: XLand-MiniGrid (Nikulin et al., 2024) and Navix (Pignatelli et al., 2024) focus on gridworld navigation, Jumanji (Bonnet et al., 2023) spans domains from simple games to NP-hard combinatorial problems, Pgx (Koyamada et al., 2023) provides classic board games, and PuzzleJAX Earle et al. (2025) enables dynamic compilation of puzzle games.

Despite this coverage, a critical gap remains: classic arcade games. While MinAtar provides simplified versions of Atari games, the full visual complexity and authentic game mechanics of classic arcade games remain absent from the JAX ecosystem. OCTAX addresses this gap by providing the first end-to-end JAX implementation of classic arcade games through CHIP-8 emulation, delivering computational benefits while preserving the engaging gameplay mechanics that made arcade games valuable for algorithmic development.

## 3 OCTAX: THE ACCELERATED CHIP-8 PLATFORM

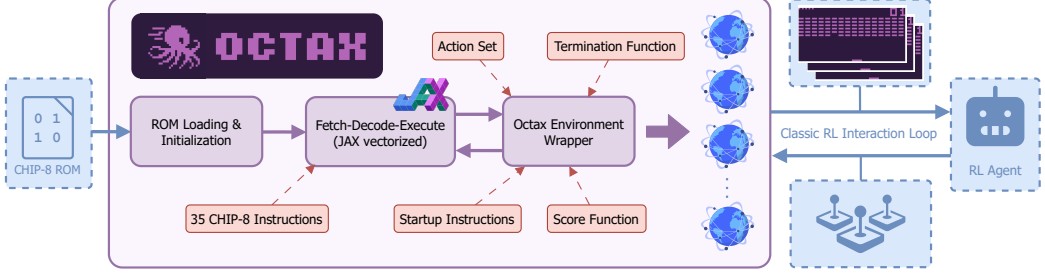

Figure 2: OCTAX architecture: ROM loading, CHIP-8 emulation pipeline, and RL environment integration. The system transforms game ROMs through fetch-decode-execute cycles into vectorized JAX operations suitable for GPU acceleration.

This section presents our JAX implementation of CHIP-8 emulation. We detail the design decisions that enable GPU acceleration while maintaining behavioral fidelity to original games, and explain how CHIP-8's architecture provides an optimal foundation for scalable experimentation in RL. Figure 2 summarizes this section.

### 3.1 WHY CHIP-8 FOR RL RESEARCH?

CHIP-8 represents a strategic choice for RL environment design. Created in the 1970s as a virtual machine specification, CHIP-8 features a 64×32 monochrome display, 16 registers, 4KB memory,

and 35-instruction set. These constraints, originally imposed by early microcomputer limitations, create several research advantages.

The platform provides image-based environments comparable to Atari games while offering some computational advantages. The 4KB memory footprint allows thousands of simultaneous game instances without memory constraints. The simple instruction set reduces emulation overhead compared to complex modern processors. The deterministic execution model ensures experimental reproducibility across different hardware configurations.

The platform supports everything from precise action games requiring split-second timing to complex puzzles demanding long-horizon planning. The 16-key input system provides sufficient complexity for interesting control challenges while remaining tractable for systematic analysis. Most importantly, CHIP-8 games are inherently modifiable and analyzable: their simple assembly code can be automatically generated, modified, and assessed for difficulty, enabling novel research directions in environment design and curriculum learning. This combination of Atari-like visual complexity with modern computational efficiency makes CHIP-8 well-suited for the JAX ecosystem, where extensive parallelization can transform week-long experiments into hour-long runs.

### 3.2    How does Octax work?

Octax converts CHIP-8 ROMs[1] into vectorized RL environments while maintaining compatibility with original games. The implementation leverages JAX's functional programming model and vectorization capabilities to enable GPU acceleration.

**ROM loading and initialization.** Game data is loaded from `.ch8` files into the emulator's 4KB memory space starting at address 0x200, following the standard CHIP-8 program layout first introduced in Weisbecker (1978). The system initializes with font data at address 0x50, sixteen general-purpose registers (V0-VF), an index register (I), a program counter (PC), and the 64×32 monochrome display buffer.

**Fetch-decode-execute cycle.** The core emulation loop implements the classic processor cycle using JAX primitives. The `fetch()` function retrieves 16-bit instructions from memory and advances the program counter. The `decode()` function extracts instruction components through bitwise operations, identifying opcodes, register indices, and immediate values. The `execute()` function uses JAX's `lax.switch` for GPU-compatible instruction dispatch to specialized handlers.

**Vectorized instruction execution.** Instruction handlers follow JAX's functional programming model, treating state as immutable and returning updated copies. ALU operations handle arithmetic and bitwise logic with carry/borrow flag management. Control flow instructions implement jumps, calls, and conditional operations using `lax.cond`. The display system uses vectorized operations to render sprites across the entire framebuffer simultaneously.

**Environment integration.** The `OctaxEnv` wrapper transforms the emulator into a standard RL interface. Each RL step executes multiple CHIP-8 instructions to maintain authentic game timing relative to the original 700Hz instruction frequency. The default frame skip setting preserves realistic game dynamics. Observations consist of the 64×32 display with 4-frame stacking, producing (4, 64, 32) boolean arrays. Actions map from discrete RL outputs to game-specific key subsets plus a no-op option. The wrapper manages delay and sound timers at 60Hz and executes startup sequences to bypass menu screens. Even though some games use the `RND` Chip-8 instruction and are therefore stochastic, we provide additional wrappers for no-op reset and sticky actions.

### 3.3    How to transform games into RL environments?

Converting CHIP-8 games into RL environments requires extracting reward signals and termination conditions from game-specific memory layouts and register usage patterns.

**Score function design.** Games store scores in different registers using various encoding schemes. Octax provides game-specific `score_fn` functions that extract scores from appropriate memory locations. Brix stores its score in register V5, incrementing with each destroyed brick. Pong encodes

---

[1]ROM stands for Read-Only Memory, a type of storage originally used in game cartridges to hold software that cannot be modified by the user.

scores in BCD format within register V14, requiring `score = (V[14] // 10) - (V[14] % 10)` to compute player advantage. Our decision to support modular rewards is intentional; however, it is incompatible with adopting human-normalized scoring schemes like those used in ALE.

**Termination logic.** Games signal completion through different register states that must be identified through analysis. Brix terminates when lives (V14) reach zero, while Tetris uses a dedicated game-state register (V1) that equals 2 on game over. Some games require compound conditions: Space Flight ends when either lives reach zero or a level completion counter exceeds a threshold, implemented as `terminated = (V[9] == 0) | (V[12] >= 0x3E)`.

**Action space optimization.** Most games use subsets of the 16-key hexadecimal keypad. OCTAX supports custom `action_set` arrays that map RL action indices to relevant keys. Pong requires only keys 1 and 4 for paddle movement, while Worm uses directional keys 2, 4, 6, 8. This reduces action space size and accelerates learning by eliminating irrelevant inputs.

**Initialization handling.** Many games include menu screens that interfere with RL training. OCTAX supports `startup_instructions` parameters that automatically execute instruction sequences during environment reset, bypassing menus to begin gameplay immediately.

We address CHIP-8's non-standardized scoring and termination by combining static ROM analysis and dynamic memory monitoring during gameplay, as detailed in Appendix C.

### 3.4 WHICH GAMES DOES OCTAX SUPPORT?

OCTAX provides a curated collection of classic CHIP-8 games across multiple genres and difficulty levels. The current implementation includes 21 titles, with additional games planned for future releases. All environments maintain full compatibility with both Gymnasium and Gymnax APIs.

| Category | Available Games | Required Capabilities |
|---|---|---|
| Puzzle | Tetris, Blinky, Worm | Long-horizon planning, spatial reasoning |
| Action | Brix, Pong, Squash, Vertical Brix, Wipe Off, Filter | Timing, prediction, reactive control |
| Strategy | Missile Command, Rocket, Submarine, Tank Battle, UFO | Resource management, tactical decisions |
| Exploration | Cavern (7 levels), Flight Runner, Space Flight (10 levels), Spacejam! | Spatial exploration, continuous navigation |
| Shooter | Airplane, Deep8, Shooting Stars | Simple reaction, basic timing |

Table 1: Currently implemented games in OCTAX.

The games (Figure 1) vary across multiple dimensions of difficulty and cognitive demand. Temporal complexity ranges from immediate reactions to long-term planning requirements. Spatial complexity spans single-screen environments to multi-screen worlds requiring navigation. Reward structures include both dense scoring mechanisms and sparse achievement-based systems. This systematic variation enables controlled studies of algorithmic performance across different challenge types while maintaining a unified technical framework for fair comparison. A categorization of these games is provided in Table 1, with more detailed descriptions available in Appendix C.3.

## 4 EXPERIMENTAL EVALUATION

We evaluate OCTAX through RL training experiments across 16 diverse CHIP-8 games. Our goal is to demonstrate that the environments present varied difficulties and learning dynamics suitable for RL research. We then evaluate the platform's computational performance.

### 4.1 HOW DO RL AGENTS LEARN IN OCTAX?

We train Proximal Policy Optimization (PPO) (Schulman et al., 2017) agents across our game suite due to its widespread adoption and proven scalability with parallel environments (Rudin et al., 2022), and Parallel Q-Network (PQN) (Gallici et al., 2024) as a modern value-based method that also benefits from parallel environments.

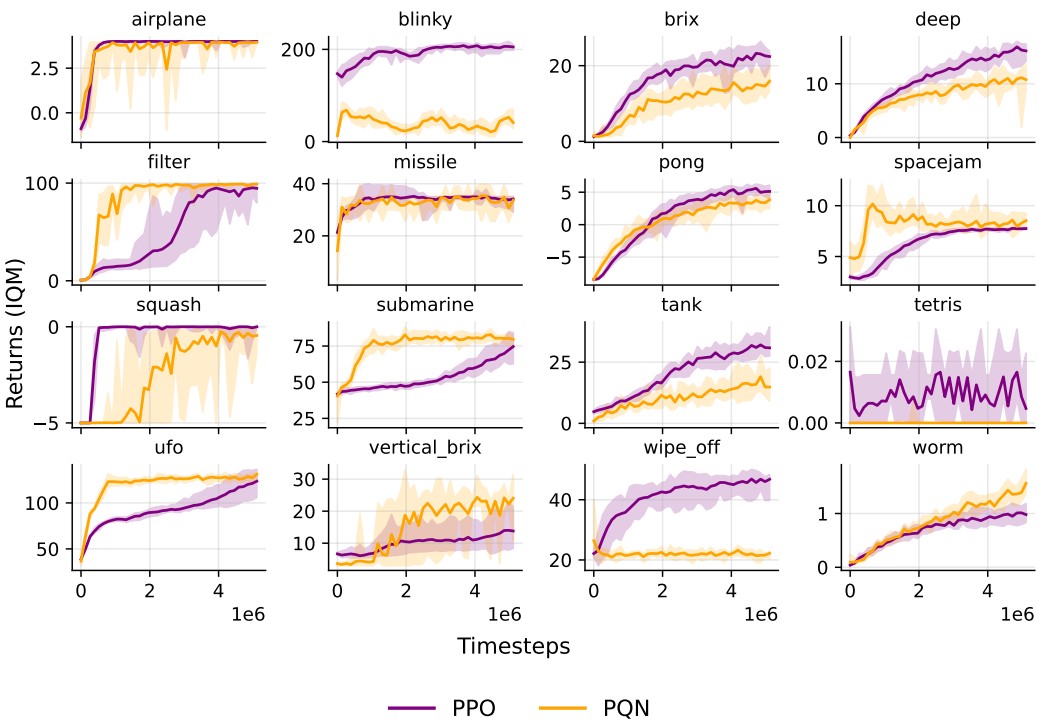

Figure 3: PPO and PQN learning curves across 16 games: interquartile mean returns using 10th-90th percentile ranges over 5M timesteps, with confidence intervals computed across 12 random seeds.

**Network architecture.** Both agents[2] uses the same architecture, for fair comparison: a convolutional neural network designed for OCTAX's (4, 64, 32) stacked observations. The feature extractor consists of three convolutional layers with 32, 64, and 64 filters, respectively. These layers use kernel sizes of (8,4), 4, and 3 with corresponding strides of (4,2), 2, and 1. Extracted features are flattened and fed to separate actor and critic heads, each containing a single 256-unit hidden layer with ReLU activation throughout (and LayerNorm for PQN).

**Training configuration.** We combine grid search optimization (detailed in Appendix D) on Pong with CleanRL's standard Atari PPO hyperparameters (Huang et al., 2022). This yields GAE lambda of 0.95, clipping epsilon of 0.2, value function coefficient of 0.5, and entropy coefficient of 0.01 for PPO. For PQN we use $\lambda = 0.9$ and an epsilon-greedy exploration decaying from 1. to 0.05 during 10% of the training. Each experiment uses 512 parallel environments with 32-step rollouts, 4 training epochs per update, and 32 minibatches for gradient computation. We apply the Adam optimizer (Kingma & Ba, 2014) with a learning rate of $5 \times 10^{-4}$ and gradient clipping to ensure stable training across 5 million timesteps per environment.

**Experimental setup.** We conduct 12 independent training runs per game using different random seeds. All experiments run on a single NVIDIA A100 GPU with 24 concurrent training sessions. Agent performance is assessed every 131,072 timesteps on 128 parallel environments.

**Results analysis.** The training curves in Figure 3 reveal distinct learning profiles across games. We observe three main patterns that reflect different cognitive demands and exploration needs. *Rapid plateau games* (Airplane, Brix, Deep, Filter, Blinky) show quick initial learning followed by stable performance, suggesting clear reward signals. *Gradual improvement games* (Pong, Tank, Vertical Brix) learn continuously over the course of training, indicating either sparser reward structures or more complex strategic requirements. *Limited performance games*, like Tetris, show little absolute

---

[2]Based on Rejax implementation (Liesen et al., 2024).

progress, making them difficult for methods without targeted exploration. Similarly, in Worm (a Snake clone), agents often manage to eat only a single apple before dying.

These learning profiles support the diversity of CHIP-8 environments, demonstrating that different games test varied aspects of learning and control. Individual training runs averaged 65 minutes each, with 24 experiments running concurrently, achieving approximately 30,800 environment steps per second across all parallel sessions.

## 4.2    HOW DOES OCTAX SCALE WITH PARALLELIZATION?

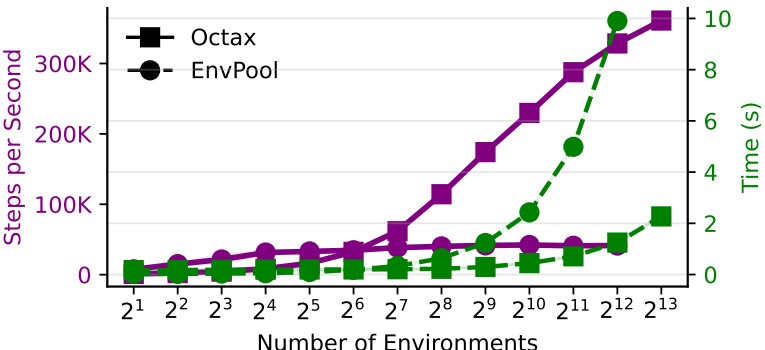

Figure 4: Performance scaling of OCTAX and EnvPool across parallelization levels. The solid purple line is the number of steps per second (higher is better), and the dashed green line is the total execution time in seconds (lower is better).

**Experimental setup.** We measure environment throughput across different parallelization levels to quantify OCTAX's computational advantages. This experiment isolates pure computational benefits by fixing the game (Pong) and agent behavior (constant action) while varying parallel environment instances. Since all environments execute identical CHIP-8 computational cycles, these performance measurements apply uniformly across the entire game suite. To better interpret our results, we compare against EnvPool because it is widely adopted in RL research, using ALE Pong to assess CPU vs. GPU-based environment scalability.

**Configuration.** We benchmark on a consumer-grade workstation with an RTX 3090 (24GB VRAM), 32GB RAM, and an Intel i7 processor (20 cores). We measure execution time for 100-step rollouts across varying parallel environment counts, with 50 independent measurements per configuration. The primary metric is environment steps per second, calculated as (number of environments × 100 steps) divided by execution time, where each step represents 4 frames due to OCTAX's default frame skip setting.

**Performance results.** Figure 4 demonstrates near-linear scaling up to 350,000 steps (or 1.4M frames) per second with 8,192 parallel environments before hitting VRAM limitations. EnvPool running ALE Pong with all available CPU cores shows reduced scaling, plateauing around 25,000 steps per second due to CPU saturation. OCTAX achieves a 14× improvement in computational efficiency at high parallelization levels, reducing the computational cost of large-scale RL experiments. We also measured GPU memory usage across different environment counts, finding that execution memory scales linearly with the number of parallel environments with our benchmark script, consuming approximately 2 MB of GPU memory per environment.

## 4.3    HOW DO LLMS ASSIST ENVIRONMENT CREATION?

Large language models (LLMs) have demonstrated a strong capability in code generation across diverse programming languages, enabling the automated creation of environments in RL research. Here we explore OCTAX's capacity to accelerate research by leveraging LLMs to generate novel

tasks, extending beyond manually designed game suites toward automated environment synthesis, as explored in Faldor et al. (2024).

**Context.** During OCTAX's development, we encountered a few games where reward and termination logic proved difficult to extract through manual analysis of game mechanics. In these cases, we decompiled ROMs to obtain CHIP-8 assembly code and successfully employed LLMs to explain the code and guide us in defining the correct `score_fn` and `terminated_fn` functions. Using `gpt-4o-mini` from the OpenAI API, we evaluated our 21 games by checking whether the model could reproduce the same score and termination functions as our hand-written implementations. While the LLM might, in principle, discover alternative reward definitions that are still semantically valid, our goal here was to assess direct match accuracy under a limited context. We found that the model performs reliably when the score is stored in a single register (57% perfect matches), but termination logic proved harder, with only 19% correct due to missed multi-register OR/AND conditions or encoded state variables. This small feasibility study illustrates that LLMs can recover simple reward signals but still require human oversight, especially with no interactive debugging or additional context. Full details and per-game analyses are provided in Appendix E. This experiment motivated us to investigate the reverse pipeline: using LLMs to generate complete CHIP-8 games from high-level descriptions, then leveraging OCTAX's scalable simulation to evaluate these procedurally created environments.

**Automated environment generation pipeline.** Our pipeline consists of seven replicable steps for automatic CHIP-8 game generation. In Step 1, we construct a corpus of CHIP-8 tutorials, documentation, and programming examples, ensuring the LLM understands the architecture's instruction set, memory layout, and common coding patterns. In Step 2, we embed this corpus into a prompt (detailed in Appendix F.1) that guides the LLM to produce syntactically correct CHIP-8 programs from high-level instructions. In Step 3, we provide a description of the game with desired mechanics, objectives, and constraints. In Step 4, the LLM generates the initial CHIP-8 code based on the provided description. In Step 5, an automated feedback loop between the LLM and a CHIP-8 compiler iteratively refines the code based on compilation errors until successful. In Step 6, Python wrapper functions for `score_fn` and `terminated_fn` are automatically generated, translating CHIP-8 registers into RL-compatible reward and termination signals. Finally, in Step 7, the game description is augmented to increase difficulty or introduce new challenges. Both the new description and the previously generated game are added to the LLM's context before next iteration. Figure 5 summarizes the automated environment generation pipeline.

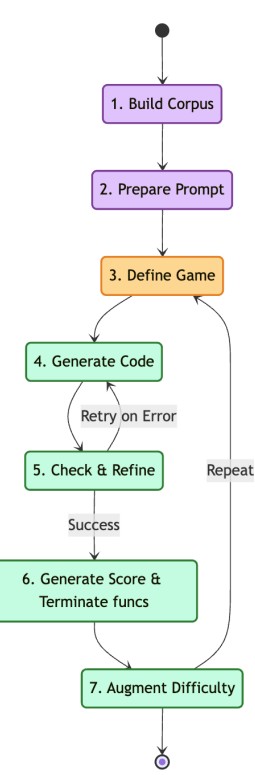

Figure 5: Environment generation pipeline.

**Target Shooter case study.** We validated this pipeline using Claude Opus 4.1, known for its proficiency in programming, with the following description: "Target Shooter – Targets appear randomly on the screen, and the player moves a crosshair to shoot them. Score increases per hit, and the game ends after a fixed number of targets." The system successfully generated three progressive difficulty levels: static targets for basic aiming skills, time-limited targets introducing decision pressure, and moving targets with time constraints requiring predictive aiming. Each level maintains consistent register mappings for score and termination, simplifying OCTAX compatibility. Figure 6 shows how the LLM-generated environment visual appearance. All the code generated by the LLM is given in F.2,

**RL experiments.** Using identical PPO configurations from Section 4.1, we trained agents on the three generated difficulty levels over 5M timesteps. Figure 7 demonstrates clear performance stratification across difficulty levels: Level 1 agents achieved optimal returns of 10.0 with rapid convergence by 1M timesteps, Level 2 agents plateaued at 9.0 returns with moderate learning speed, while Level 3 agents reached 8.0 returns with the slowest progression. The inverse relationship between difficulty level and both final performance and sample efficiency indicates that our LLM-generated environments successfully create a meaningful difficulty gradient. This proof-of-concept

demonstrates the feasibility of automated environment generation for RL research via OCTAX, with promising applications in curriculum learning, open-endedness, and continual learning scenarios.

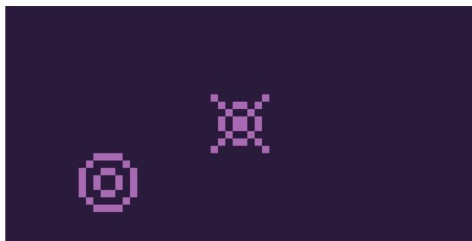

Figure 6: Rendering of the Target Shooter game showing the player (left, circular object) and target (right, cross-shaped object).

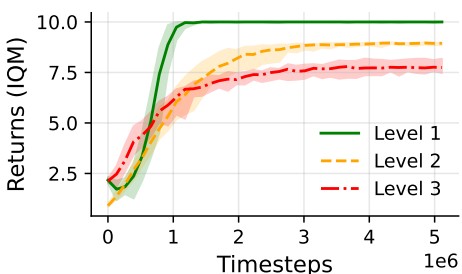

Figure 7: PPO training performance on generated environments with varying difficulty.

## 5 CONCLUSION

We introduced OCTAX, a JAX-based CHIP-8 emulation platform that provides GPU-accelerated arcade game environments for reinforcement learning research. Our implementation achieves significant performance improvements over CPU-based alternatives, enabling experiments with thousands of parallel environments while maintaining perfect behavioral fidelity to original games. Through PPO evaluation across 16 diverse games, we demonstrated varied learning dynamics that highlight the cognitive diversity within classic arcade environments. The platform's modular design enables both manual game integration and automated environment generation using large language models, providing researchers with flexible experimental design options.

**Societal and environmental impact.** OCTAX enables more rigorous evaluation with larger sample sizes, addressing reproducibility concerns that affect institutions with limited computational resources. This implementation can reduce energy consumption compared to resource-intensive benchmarks such as ALE: experiments that once required top-tier clusters can now run efficiently on a single GPU, potentially saving significant compute time and resources.

**Limitations.** The GPU-based architecture faces performance constraints due to CHIP-8's variable instruction execution complexity. JAX synchronization across parallel environments means each step's execution time depends on the slowest instruction among CHIP-8's 35 operations, typically display rendering or complex ALU operations. The absence of established maximum scores across our game suite prevents the assessment of whether agents approach theoretical performance limits, limiting evaluation of algorithmic performance ceilings.

**Future work.** OCTAX can expand through community contributions, with hundreds of compatible ROMs available online. The LLM-assisted environment generation pipeline enables curriculum learning and open-ended research through procedurally generated games that provide task diversity. We plan to investigate emulator optimizations including instruction-level parallelization strategies and adaptive batching to address synchronization bottlenecks from variable execution times. We also aim to extend platform support to Super-CHIP8 and XO-CHIP variants: Super-CHIP8 offers higher resolution displays (128×64) and extended instruction sets originally developed for HP48 calculators, while XO-CHIP provides color graphics, improved audio, and expanded memory while maintaining backward compatibility. These extensions would enable OCTAX to support more sophisticated games and visual complexity while preserving the computational efficiency advantages of the JAX-native architecture. Many CHIP-8 games feature multi-agent or multi-player mechanics, which we plan to support in future platform releases. The platform's high-throughput capabilities also position it well for offline RL research, enabling the efficient creation of large-scale datasets and the comprehensive evaluation of offline algorithms across diverse game environments.

## ACKNOWLEDGMENTS

The authors Waris Radji and Thomas Michel are affiliated with the Inria Scool team project. This work has been supported by the French Ministry of Higher Education and Research, the Hauts-de-France region, Inria, and the MEL. Additional support was provided by the French National Research Agency under the PEPR IA FOUNDRY project (ANR-23-PEIA-0003). Experiments presented in this paper were carried out using the PlaFRIM experimental testbed, supported by Inria, CNRS (LABRI and IMB), Université de Bordeaux, Bordeaux INP and Conseil Régional d'Aquitaine (see https://www.plafrim.fr).

## REPRODUCIBILITY STATEMENT

We provide complete resources to ensure reproducibility of our results. The OCTAX source code, including all 21 game environment implementations, JAX-based CHIP-8 emulator, and training scripts, is available as supplementary material. Our experimental setup uses standard PPO hyperparameters detailed in Section 4.1, with hardware specifications and performance benchmarking configurations provided in Section 4.2. All training experiments use identical network architectures and hyperparameters across games, enabling direct replication of our learning curves in Figure 3. For the LLM-assisted environment generation pipeline in Section 4.3, we include the prompt templates and generated CHIP-8 assembly code in Appendix F. The modular design of OCTAX allows researchers to extend our game suite using the technical specifications in Section 3. The repository containing all source code, experiments, and data is available at: https://github.com/riiswa/octax/.

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

## A USE OF LARGE LANGUAGE MODELS

We used Large Language Models in three capacities during this research. First, Claude Opus 4.1 serves as a core research component in Section 4.3, where we demonstrate automated CHIP-8 game generation from high-level descriptions. This represents a novel research contribution, with all generated code validated through compilation and RL experiments. Second, we employed Claude Sonnet 4 for writing assistance, including text refinement, rephrasing technical concepts, and improving academic tone. Third, LLMs generated code documentation, docstrings, and tutorial content. All research ideas, experimental design, and scientific claims originate from the authors. We did not use LLMs for ideation, hypothesis formation, or result interpretation. We manually reviewed and validated all LLM-assisted content for accuracy and take full responsibility for all presented content.

## B CHIP-8 TECHNICAL SPECIFICATIONS

### B.1 PLATFORM OVERVIEW

CHIP-8 was created by Joseph Weisbecker at RCA in the mid-1970s as a virtual machine for early microcomputers. The platform established one of the first successful portable gaming ecosystems by providing a hardware abstraction layer that enabled games to run across different systems.

### B.2 SYSTEM ARCHITECTURE

The CHIP-8 architecture consists of:

- **Memory:** 4KB total, with programs loaded at address 0x200
- **Registers:** 16 8-bit registers (V0-VF), with VF serving as a flag register
- **Display:** 64×32 pixel monochrome screen with XOR-based rendering

- **Input:** 16-key hexadecimal keypad (0-9, A-F)
- **Timers:** 60Hz delay timer and sound timer
- **Audio:** Single-tone buzzer

## B.3 INSTRUCTION SET HIGHLIGHTS

CHIP-8's 35-instruction set includes specialized gaming primitives:

- **Sprite Drawing (DXYN):** XOR-based rendering enabling collision detection
- **Key Input (EX9E, EXA1):** Skip instructions based on key state
- **BCD Conversion (FX33):** Convert register values to decimal display
- **Memory Operations:** Bulk register loading/storing (FX55, FX65)

The XOR-based sprite system is particularly elegant: drawing the same sprite twice erases it, enabling simple animation and automatic collision detection when pixels turn off.

## B.4 FONT SYSTEM

CHIP-8 includes built-in 4×5 pixel font data for hexadecimal digits (0-F), stored at addresses 0x050-0x09F. Games reference these fonts for score and text display by setting the index register to the appropriate font location.

## C GAME ENVIRONMENT IMPLEMENTATION DETAILS

### C.1 SCORE DETECTION METHODOLOGY

CHIP-8 games store scoring information in arbitrary memory locations using game-specific formats. Our automated detection operates in two phases:

**Static Analysis:** We analyze ROM structure for common programming patterns, particularly binary-coded decimal (BCD) operations (FX33 instruction) that suggest numeric display routines.

**Dynamic Monitoring:** During human gameplay sessions, we monitor memory changes to correlate locations with scoring events. Register trend analysis identifies increasing values (likely scores) versus decreasing values (likely lives/health).

### C.2 REWARD DESIGN

Each environment implements a reward function that extracts scoring information from the emulator state. The reward is computed by reading specific CHIP-8 registers (V0-VF) that track game-relevant metrics. Most environments use direct score extraction where the reward equals the value stored in a score register. Some environments apply transformations to the raw register values to shape the reward signal for reinforcement learning.

### C.3 GAME LIST

#### C.3.1 LONG-HORIZON PLANNING & SPATIAL REASONING

*Requires strategic thinking, spatial awareness, and multi-step planning*

- **tetris** – Tetris by Fran Dachille (1991): Classic Tetris with piece rotation, movement, and dropping. Uses keys 4 (rotate left), 5 (move left), 6 (move right), and 7 (drop). Speed increases every 5 lines and peaks at 45 lines. Reward: Register V[10] containing the score. Termination: Game over when V[1] equals 2 (Board overflow).
- **blinky** – Blinky by Hans Christian Egeberg (1991): Pac-Man clone where the player navigates a maze to collect pills while avoiding two ghosts (Packlett and Heward). The maze

contains one gateway and four energy pills near corners. Points awarded for each pill, energy pill, catching ghosts, and finishing the maze. Player has 2 lives. Uses keys 3, 6, 7, and 8 for movement. Reward: Register V[6] containing the score. Termination: Game over when V[3] equals 255 (Collision with ghost).

- **worm** – SuperWorm V4 by RB-Revival Studios (2007): Snake-like game for Chip8. The player navigate the worm to collect items while avoiding walls and self-collision. Uses keys 2, 8, 4, and 6 for directional movement. Reward: Register V[5] containing the score. Termination: Game over when V[7] equals 255 (Collision).

### C.3.2 TIMING, PREDICTION & REACTIVE CONTROL

*Requires precise timing, trajectory prediction, and fast reactive responses*

- **brix** – Brix by Andreas Gustafsson (1990): Breakout clone where the player controls a paddle to bounce a ball and destroy bricks. Player has 5 lives. Uses keys 4 (left) and 6 (right) for paddle movement. Reward: Register V[5] increments per brick destroyed. Termination: Game over when V[14] equals 4 (No more lives).
- **pong** – Pong: Single player pong game where the player controls a paddle to hit the ball. Uses keys 1 and 4 for paddle movement. Reward: Computed as (V[14] // 10) - (V[14] % 10), representing player score minus opponent score. Termination: Game over when either score reaches 9.
- **squash** – Squash by David Winter (1997): Bounce a ball around a squash court with paddle control. Uses keys 1 (up) and 4 (down) for paddle movement. Player has 5 lives. Reward: Register V[11] (Number of lives left). Termination: Game over when V[11] equals 0 (No more lives).
- **vertical_brix** – Vertical Brix by Paul Robson (1996): Breakout variant with vertical brick layout and paddle movement. In the original game you need to press 7 to start, but we skip that in the environment. Uses keys 1 and 4 to move the paddle vertically. Reward: Register V[8] (bricks eliminated). Termination: Game over when V[7] (remaining lives) equals 0.
- **wipe_off** – Wipe Off by Joseph Weisbecker: Move paddle left or right to wipe out spots on screen. Each spot counts 1 point. Player gets 20 balls. Uses keys 4 (left) and 6 (right) for paddle movement. Reward: Register V[6] (Number of points wiped out). Termination: Game over when V[7] (balls left) equals 0.
- **filter** – Filter: Catch drops falling from a pipe at the top of the screen with paddle. Uses keys 4 (left) and 6 (right) for paddle movement. 7 lives. Reward: Register V[14] (number of drops caught). Termination: Game over when V[13] (remaining lives) equals 0.

### C.3.3 RESOURCE MANAGEMENT & TACTICAL DECISIONS

*Requires managing limited resources and making strategic tactical choices*

- **missile** – Missile Command by David Winter (1996): Shoot 8 targets on screen using key 8. The shooter moves faster with each shot. Player has 12 missiles total and earns 5 points per target hit. Reward: Register V[7] (points). Termination: Game over when V[6] (missiles left) equals 0.
- **rocket** – Rocket by Joseph Weisbecker (1978): An enemy UFO moves from left to right across the top of the screen. Launch rockets vertically by pressing key F (15). Rockets appear at random horizontal positions at the bottom. Score increments by 1 when UFO is hit. Player has 9 rockets total. Reward: Register V[1] (points). Termination: Game over when V[2] (rocket launched) equals 9.
- **submarine** – Submarine by Carmelo Cortez (1978): Fire depth charges at submarines below using key 5. Score 15 points for hitting a small submarine and 5 points for a large submarine. Player starts with 25 depth charges. Reward: Register V[7] (points). Termination: Game over when V[8] (remaining lives) equals 0.
- **tank** – Tank Battle: Control a tank with 25 bombs to hit a mobile target. Uses keys 2, 4, 5, 6, and 8 to move. If the tank hits the target, player loses 5 bombs. Reward: Register V[14] (points). Termination: Game over when V[6] (bombs left) equals 0.

- **ufo** – UFO by Lutz V (1992): Stationary missile launcher at the bottom of the screen shoots at flying objects. Uses keys 4 (left diagonal), 5 (straight up), and 6 (right diagonal) to fire. Player has 15 missiles. Score displayed on left, remaining missiles on right. Reward: Register V[7] (score). Termination: Game over when V[8] (missiles left) equals 0.

### C.3.4   EXPLORATION & CONTINUOUS NAVIGATION

*Requires spatial exploration, obstacle avoidance, and continuous movement control*

- **cavern** – Cavern by Matthew Mikolay (2014): Navigate through a cave without crashing into walls. Uses keys 2, 4, 6, and 8 for movement. Modified ROM with leftward progress reward system where V[0] increments for each new leftmost X position reached, encouraging leftward exploration. V[A] tracks leftmost position ever visited. Reward: Register V[0]. Termination: Game over when V[14] equals 0 (crash into wall).
- **flight_runner** – Flight Runner by TodPunk (2014): Simple flight navigation game. Uses keys 5, 7, 8, and 9 for movement controls. Reward: Register V[7]. Termination: Game over when V[5] or V[7] equals 255.
- **space_flight** – Space Flight: Fly through an asteroid field from left to right avoiding obstacles. Uses keys 1 and 4 to navigate the spaceship. Modified ROM with single life mode and immediate termination on collision. V[0] increments per frame survived as distance score. V[9] tracks lives. Reward: Register V[0]. Termination: Game over when V[9] equals 0 or V[12] is at least 0x3E.
- **spacejam** – Spacejam! by William Donnelly (2015): Ship tunnel navigation game based on ShipTunnel from 2014 OctoJam. Uses keys 5, 8, 7, and 9 for movement. Reward: Register V[9]. Termination: Game over when V[10] equals 0 (Ship destroyed).

### C.3.5   SIMPLE REACTION & TIMING

*Requires basic reaction time and simple decision making*

- **airplane** – Airplane: Bombing game where bombs are dropped by pressing key 8. V[11] tracks remaining targets and V[12] tracks level progression. Reward: Computed as -V[11] - V[12], rewarding target hits and penalizing level progression. Termination: Game over when V[11] equals 0 or V[12] equals 6.
- **deep** – Deep8 by John Earnest (2014): Move boat left and right with keys 7 and 9. Press key 8 to drop a bomb and release to detonate it. Destroy incoming squid before they tip the boat. Reward: Register V[9]. Termination: Game over when V[B] does not equal 1.
- **shooting_stars** – Shooting Stars by Philip Baltzer (1978): Classic shooting game. Uses keys 2, 8, 4, and 6 for movement. Reward: Register V[0] with capping at 128 (returns 0 if V[0] exceeds 128). Termination: Never terminates.

## D   HYPERPARAMETER OPTIMIZATION RESULTS

We conducted a comprehensive grid search on the Pong environment to identify optimal PPO hyperparameters before evaluating across the full game suite. The search explored four key dimensions: number of parallel environments, rollout length, minibatch size, and learning rate. All experiments used 4 epochs per update, GAE lambda of 0.95, and gradient clipping at 0.5.

### D.1   SEARCH SPACE

The hyperparameter search explored the following ranges:

- **Environments**: $\{128, 256, 512, 1024\}$
- **Rollout steps**: $\{32, 64, 128, 512\}$
- **Minibatches**: $\{4, 8, 16, 32\}$
- **Learning rate**: $\{2.5 \times 10^{-4}, 5 \times 10^{-4}, 1 \times 10^{-3}\}$

Each configuration was trained for 1M timesteps with evaluation every 65,536 steps. Final evaluation scores represent the last recorded performance, where less negative values indicate better performance.

## D.2 RESULTS SUMMARY

Table 2 presents the key configurations and their final evaluation scores. Higher scores indicate better performance (scores are negative, with values closer to zero being better).

Table 2: Hyperparameter search results on Pong environment. Configurations sorted by final evaluation score.

| Envs | Steps | Minibatches | LR | Score |
|------|-------|-------------|---------|-------|
| 512 | 32 | 32 | 0.0005 | -2.34 |
| 512 | 32 | 16 | 0.001 | -2.48 |
| 512 | 32 | 32 | 0.001 | -2.69 |
| 128 | 128 | 16 | 0.00025 | -2.95 |
| 128 | 64 | 8 | 0.00025 | -3.19 |
| 512 | 32 | 16 | 0.00025 | -3.20 |
| 256 | 64 | 16 | 0.00025 | -3.38 |
| 128 | 32 | 4 | 0.00025 | -3.44 |
| 128 | 64 | 16 | 0.00025 | -3.53 |
| 512 | 32 | 16 | 0.0005 | -3.73 |
| 256 | 32 | 4 | 0.00025 | -3.78 |
| 128 | 128 | 8 | 0.00025 | -3.91 |
| 256 | 128 | 32 | 0.00025 | -4.03 |
| 512 | 64 | 16 | 0.00025 | -4.17 |
| 1024 | 32 | 32 | 0.00025 | -4.34 |
| 1024 | 32 | 16 | 0.00025 | -4.44 |
| 256 | 128 | 16 | 0.00025 | -4.66 |
| 1024 | 64 | 32 | 0.00025 | -4.96 |

## D.3 ANALYSIS AND KEY FINDINGS

**Learning rate impact.** Higher learning rates significantly improved performance, with $5 \times 10^{-4}$ and $1 \times 10^{-3}$ substantially outperforming $2.5 \times 10^{-4}$. The top three configurations all used learning rates above the commonly used $2.5 \times 10^{-4}$.

**Environment scaling.** 512 parallel environments provided the optimal balance between computational efficiency and sample diversity. Configurations with 1024 environments showed diminishing returns, possibly due to computational overhead or reduced gradient update frequency.

**Rollout length.** Shorter rollouts (32 steps) consistently outperformed longer ones, indicating more frequent policy updates may be beneficial for this environment.

**Minibatch size.** Larger minibatch sizes (16-32) generally improved performance by providing more stable gradient estimates, though the effect was less pronounced than learning rate changes.

## D.4 FINAL CONFIGURATION

Based on these results, we selected the following hyperparameters for all subsequent experiments:

- Parallel environments: 512
- Rollout steps: 32
- Training epochs: 4
- Minibatches: 32
- Learning rate: $5 \times 10^{-4}$
- GAE lambda: 0.95
- Clip epsilon: 0.2

- Value function coefficient: 0.5
- Entropy coefficient: 0.01

# E    EVALUATION STUDY ON LLM-GENERATED REWARD FUNCTIONS

We evaluate whether LLMs can extract reward and termination functions from raw CHIP-8 assembly code, comparing LLM outputs against our human-defined implementations. Given CHIP-8 assembly code without symbolic information and any documentation, the LLM must implement two functions using only 16 registers: `score_fn(state)` to extract the current game score, and `terminated_fn(state)` to determine if the game has ended.

*Important caveat*: Exact register matches indicate correct understanding of the original implementation, but alternative register choices may still represent valid game semantics.

We used the model `gpt-4o-mini` from the OpenAI API due to its low price. The full prompt we use is shown below:

```
You are analyzing CHIP-8 assembly code to extract reward and
termination logic for reinforcement learning.

CHIP-8 ASSEMBLY CODE for '{rom_name}':
{game_code}

TASK: Implement score_fn() and terminated_fn() using ONLY
the 16 registers in state.V[0-15].

STEP-BY-STEP ANALYSIS REQUIRED:

1. REGISTER IDENTIFICATION:
   - Scan the assembly for registers that track score, lives,
     or game state
   - Look for patterns: incrementing (score), decrementing (lives),
     flag checks (game over)
   - Note: Multiple registers may be involved (e.g., V[9] AND V[12])

2. SCORE FUNCTION:
   - Which register(s) hold the score value?
   - Is it a simple read, or does it require calculation
     (e.g., BCD decode, multi-register)?
   - Provide evidence from assembly code

3. TERMINATION FUNCTION:
   - Which register(s) indicate game over?
   - What are the exact conditions? (equal, not equal, AND, OR?)
   - Check for: lives==0, game_state==X, score>=limit, etc.
   - Provide evidence from assembly code

OUTPUT FORMAT:
# Game: [One-line description]
# ANALYSIS:
# Score register(s): V[X] because [reason from code]
# Termination register(s): V[Y] because [reason from code]

def score_fn(state): return state.V[X]
def terminated_fn(state): return state.V[Y] == Z

CRITICAL REMINDERS:
- Look for ALL conditions in termination (often OR/AND combinations)
- Verify register choices against actual assembly operations
```

```
- Don't guess - justify each register choice with code evidence
```

## E.1 EVALUATION

We classify LLM outputs into four categories. **Perfect**: exact register(s) and logic match ground truth; **Partial**: correct register(s) but incomplete/incorrect logic; **Wrong Logic**: uses ground truth registers in wrong context; **Failure**: completely unrelated register(s) to ground truth. Results are summarized in Table 3.

Table 3: LLM Function Generation Results (21 CHIP-8 Games)

| Game | Score Function | Termination Function |
|---|---|---|
| Airplane | Partial | Failure |
| Blinky | Failure | Failure |
| Brix | Perfect | Failure |
| Cavern | Perfect | Perfect |
| Deep8 | Perfect | Wrong Logic |
| Filter | Perfect | Perfect |
| FlightRunner | Perfect | Failure |
| Missile | Perfect | Partial |
| Pong | Partial | Failure |
| Rocket | Perfect | Failure |
| Shooting Stars | Partial | Failure |
| SpaceFlight | Wrong Logic | Failure |
| Spacejam | Perfect | Failure |
| Squash | Perfect | Wrong Logic |
| Submarine | Perfect | Perfect |
| Tank | Failure | Failure |
| Tetris | Failure | Failure |
| UFO | Perfect | Perfect |
| VerticalBrix | Wrong Logic | Failure |
| WipeOff | Perfect | Failure |
| Worm | Failure | Failure |
| **Perfect** | **12/21 (57%)** | **4/21 (19%)** |
| **Partial** | **3/21 (14%)** | **1/21 (5%)** |
| **Wrong Logic** | **3/21 (14%)** | **2/21 (10%)** |
| **Failure** | **3/21 (14%)** | **14/21 (67%)** |

**Perfect matches: simple single-register cases.** The LLM succeeded when game logic used trivial register assignments. In Submarine, both functions were correctly identified because the assembly showed clear `v7 += 0x05` for scoring and `if v8 == 0x00` for game-over checks—unambiguous patterns that the LLM easily recognized.

**Partial success: correct registers, wrong logic.** In Pong, the LLM identified V[14] as the score register but missed Binary-Coded Decimal (BCD) encoding where 0x23 represents player=2, opponent=3. The ground truth decodes this as `(V[14]//10)-(V[14]%10)` while the LLM simply returned `V[14]`. Similarly, in Shooting Stars, the LLM missed overflow protection: the ground truth returns 0 when V[0] exceeds 128, but the LLM returned raw V[0]. These cases show the LLM can identify score registers but struggles with encoding schemes and boundary conditions.

**Wrong logic: register confusion within context.** In Squash, the LLM confused adjacent registers: it correctly identified V[11] for scoring but used V[12] for termination when V[11] actually serves dual purpose (score and lives). This "off-by-one" pattern appeared repeatedly (Missile: V[5] vs V[6], Tank: V[6] vs V[13]). Both registers appear in the assembly's game logic, but the LLM incorrectly assumed separate registers for each function rather than recognizing dual-purpose usage.

**Complete failures: unrelated register selection.** In Blinky, the LLM chose entirely wrong registers (V[9] for score vs ground truth V[6], and V[6]/V[7] for termination vs ground truth V[3]). Analysis of the LLM's reasoning revealed it focused on sprite drawing operations involving these registers rather than actual score-tracking logic

**Termination logic: the major challenge** Termination functions achieved only 19% perfect accuracy, primarily due to multi-condition logic. In SpaceFlight, the game ends when either lives (V[9]) reach zero OR distance (V[12]) exceeds threshold: `(V[9]==0)|(V[12]>=0x3E)`. The LLM output only `V[14]==0`—a single condition using an unrelated register. This pattern repeated: the LLM rarely captured compound conditions with multiple registers. In Airplane, the LLM detected that V[12] reaches 6 at game end but used wrong register V[10] and completely missed the V[11] lives condition. Even with explicit prompt instructions to check for OR/AND combinations, the LLM showed strong bias toward single-condition checks.

### E.2 DISCUSSION AND IMPLICATIONS

**Score functions** achieved 57% perfect accuracy, with 71% correctly identifying the primary register. Success correlated with simple increment patterns (`v7 += 0x01`) and direct reads. Failures occurred with multi-register calculations, encoding schemes like BCD, and negative scores requiring subtraction. **Termination functions** struggled at 19% accuracy with 67% failures, stemming from multi-condition OR/AND logic, multiple register interactions, and flag-based state machines.

**Practical implications**: LLMs can assist with reward function extraction for simple game mechanics but require human verification for games with multiple termination conditions, complex scoring systems, or multi-register state dependencies.

**Limitations**: Our evaluation measures implementation matching, not semantic equivalence. Alternative register choices may provide valid rewards for RL training even when differing from human implementations.

## F LLM-ASSISTED ENVIRONMENT GENERATION

This appendix details the automated environment generation pipeline using large language models (LLMs) to create novel CHIP-8 games for reinforcement learning research. We demonstrate the complete process from prompt engineering to code generation across three difficulty levels of a Target Shooter game.

### F.1 PROMPT ENGINEERING

Our LLM generation pipeline relies on carefully crafted prompts that provide comprehensive CHIP-8 programming context and specific game requirements. The core prompt structure includes CHIP-8 architectural constraints, Octo assembly language syntax, and reinforcement learning compatibility requirements.

Listing 1: LLM prompt template for CHIP-8 game generation

```
You are a **professional CHIP-8 (classic version) game developer**.
Your task is to **design and implement new CHIP-8 games in Octo assembly
    language**. I will provide you with tutorials and references for Octo
     assembly. You must be rigorous and ensure that your code is **
    syntactically correct, runnable, and follows CHIP-8 conventions**.

<documentation></documentation>

<tutorial1></tutorial1>

<tutorial2></tutorial2>

<example></example>
```

```
The goal is to create a **game suitable for reinforcement learning (RL)
    research**, which means:
* The **score** must be stored in a clear and consistent register or
    memory location.
* The **termination condition** (game over) must also be easily
    extractable (e.g., through a specific flag or register value).
* The game should have **deterministic rules** and be lightweight enough
    for training agents.

Here is the description of the game you must implement:

<description>{{description}}</description>
```

The prompt incorporates several key components:

- **Role specification**: Establishes the LLM as a professional CHIP-8 developer
- **Technical constraints**: Emphasizes syntactic correctness and CHIP-8 compliance
- **RL compatibility**: Specifies requirements for score tracking and termination detection
- **Reference material**: Includes comprehensive CHIP-8 documentation and examples
- **Game description**: Placeholder for specific game mechanics and objectives

The prompt template includes placeholder tags that are populated with comprehensive CHIP-8 resources: <documentation> contains the official Octo Manual (https://johnearnest. github.io/Octo/docs/Manual.html), <tutorial1> includes the Beginner's Guide (https://johnearnest.github.io/Octo/docs/BeginnersGuide.html), <tutorial2> incorporates the Intermediate Guide (https://johnearnest.github. io/Octo/docs/IntermediateGuide.html), and <example> provides a complete game implementation (https://github.com/JohnEarnest/chip8Archive/blob/ master/src/outlaw/outlaw.8o) to demonstrate best practices and coding patterns.

### F.2    GENERATED TARGET SHOOTER IMPLEMENTATION

Using the prompt template, we generated three progressive difficulty levels of a Target Shooter game. Each level maintains consistent register mappings for score and termination while introducing increasing complexity in target behavior and timing constraints.

#### F.2.1    LEVEL 1: STATIC TARGETS

The first difficulty level features stationary targets that appear at random locations, focusing on basic aiming and shooting mechanics.

Listing 2: Level 1 Target Shooter - Static targets

```
################################################
#
#  Target Shooter - RL Training Game
#
#  A deterministic shooting game designed for
#  reinforcement learning research.
#
#  Controls:
#  - WASD to move crosshair
#  - E to shoot
#
#  Score is stored in register v2 (score_reg)
#  Game over flag in register v3 (gameover_reg)
#
#  Game ends after hitting 10 targets.
#
################################################
```

```
# Sprite data
: crosshair
        0b10000001
        0b01011010
        0b00100100
        0b01011010
        0b01011010
        0b00100100
        0b01011010
        0b10000001

: target
  0b00111100
        0b01000010
        0b10011001
        0b10100101
        0b10100101
        0b10011001
        0b01000010
        0b00111100

##################################################
#  Register Map - Critical for RL extraction
##################################################

:alias crosshair_x    v0  # Crosshair X position
:alias crosshair_y    v1  # Crosshair Y position
:alias score_reg      v2  # SCORE - RL agents read this!
:alias gameover_reg   v3  # GAME OVER FLAG (0=playing, 1=over)
:alias target_x       v4  # Target X position
:alias target_y       v5  # Target Y position
:alias target_active  v6  # Target active flag
:alias temp1          v7  # Temporary register
:alias temp2          v8  # Temporary register
:alias shot_active    v9  # Shot in progress flag
:alias targets_hit    va  # Count of targets hit (max 10)
:alias key_reg        vb  # Key input register

:const MAX_TARGETS 10     # Game ends after 10 targets
:const TARGET_SIZE 8      # Target sprite size
:const CROSSHAIR_SIZE 8   # Crosshair sprite size
:const POINTS_PER_HIT 1   # Points awarded per target hit

##################################################
#  Main Game Entry Point
##################################################

: main
  # Initialize game state
  score_reg     := 0   # Score starts at 0
  gameover_reg  := 0   # Game is not over
  targets_hit   := 0   # No targets hit yet
  target_active := 0   # No target active initially
  shot_active   := 0   # No shot in progress

  # Initial crosshair position (center)
  crosshair_x := 28
  crosshair_y := 12

  clear

  # Draw initial UI
  draw-crosshair
```

```
  # Main game loop
  loop
    # Check if game should end
    if targets_hit == MAX_TARGETS then jump game-over

    # Spawn new target if none active
    if target_active == 0 then spawn-target

    # Handle player input
    handle-input

    # Check for hit if shot was fired
    if shot_active == 1 then check-hit

    # Small delay for playability
    temp1 := 1
    delay := temp1
    wait-delay

  again

##################################################
#  Game Over Handler
##################################################

: game-over
  gameover_reg := 1    # Set game over flag for RL agent

  # Flash screen to indicate game over
  temp1 := 0
  loop
    clear
    temp2 := 5
    delay := temp2
    wait-delay

    draw-crosshair
    if target_active == 1 then draw-target
    temp2 := 5
    delay := temp2
    wait-delay

    temp1 += 1
    if temp1 != 3 then
  again

  # Infinite loop - game is over
  loop
    # RL agent should detect gameover_reg == 1
  again

##################################################
#  Input Handling
##################################################

: handle-input
  # Save current position
  temp1 := crosshair_x
  temp2 := crosshair_y

  # Movement controls (WASD) - use consistent key codes
  key_reg := 7  # A key (left)
  if key_reg key then temp1 += -2

  key_reg := 9  # D key (right)
```

```
  if key_reg key then temp1 += 2

  key_reg := 5  # W key (up)
  if key_reg key then temp2 += -2

  key_reg := 8  # S key (down)
  if key_reg key then temp2 += 2

  # Boundary checking
  if temp1 >= 254 then temp1 := 0   # Left boundary (wrapping check)
  if temp1 >= 56 then temp1 := 56   # Right boundary
  if temp2 >= 254 then temp2 := 0   # Top boundary (wrapping check)
  if temp2 >= 24 then temp2 := 24   # Bottom boundary

  # Check if position changed
  if temp1 != crosshair_x then jump update-crosshair
  if temp2 != crosshair_y then jump update-crosshair

  # Check for shoot (E key)
  key_reg := 6
  if key_reg key then shot_active := 1

  return

: update-crosshair
  # Erase old crosshair
  i := crosshair
  sprite crosshair_x crosshair_y CROSSHAIR_SIZE

  # Update position
  crosshair_x := temp1
  crosshair_y := temp2

  # Draw new crosshair
  sprite crosshair_x crosshair_y CROSSHAIR_SIZE

  # Check shoot after movement
  key_reg := 6
  if key_reg key then shot_active := 1
;

#################################################
#   Target Management
#################################################

: spawn-target
  # Generate random position for target
  target_x := random 0x37  # 0-55 range
  target_y := random 0x17  # 0-23 range

  # Ensure minimum distance from edges
  if target_x <= 2 then target_x := 3
  if target_y <= 2 then target_y := 3

  target_active := 1
  draw-target
;

: draw-target
  i := target
  sprite target_x target_y TARGET_SIZE
;

: draw-crosshair
  i := crosshair
```

```
    sprite crosshair_x crosshair_y CROSSHAIR_SIZE
;

#################################################
#   Hit Detection
#################################################

: check-hit
  shot_active := 0  # Reset shot flag

  # Check if target is active
  if target_active == 0 then return

  # Simple hit detection - check if crosshair center is near target
      center
  # Calculate X distance
  temp1 := crosshair_x
  temp1 += 4  # Crosshair center
  temp2 := target_x
  temp2 += 4  # Target center

  # Check X proximity
  if temp1 > temp2 then jump check-x-greater

  # crosshair is left of or at target
  temp2 -= temp1
  if temp2 > 6 then return  # Too far
  jump check-y-axis

: check-x-greater
  # crosshair is right of target
  temp1 -= temp2
  if temp1 > 6 then return  # Too far

: check-y-axis
  # Calculate Y distance
  temp1 := crosshair_y
  temp1 += 4  # Crosshair center
  temp2 := target_y
  temp2 += 4  # Target center

  # Check Y proximity
  if temp1 > temp2 then jump check-y-greater

  # crosshair is above or at target
  temp2 -= temp1
  if temp2 > 6 then return  # Too far
  jump register-hit

: check-y-greater
  # crosshair is below target
  temp1 -= temp2
  if temp1 > 6 then return  # Too far

: register-hit
  # Hit confirmed!
  # Erase target
  draw-target
  target_active := 0

  # Update score (for RL agent)
  score_reg += POINTS_PER_HIT
  targets_hit += 1

  # Sound feedback
```

```
  temp1 := 3
  buzzer := temp1
;

#################################################
#  Utility Functions
#################################################

: wait-delay
  loop
    temp1 := delay
    if temp1 != 0 then
  again
;
```

### F.2.2 LEVEL 2: TIME-LIMITED TARGETS

The second level introduces time pressure by making targets disappear after a fixed duration, requiring faster decision-making from RL agents.

Listing 3: Level 2 Target Shooter - Time-limited targets

```
#################################################
#
#  Target Shooter - RL Training Game
#
#  A deterministic shooting game designed for
#  reinforcement learning research.
#
#  Controls:
#  - WASD to move crosshair
#  - E to shoot
#
#  Score is stored in register v2 (score_reg)
#  Game over flag in register v3 (gameover_reg)
#
#  Game ends after 10 targets (hit or missed).
#  Targets disappear after ~3 seconds if not hit.
#
#################################################

# Sprite data
: crosshair
        0b10000001
        0b01011010
        0b00100100
        0b01011010
        0b01011010
        0b00100100
        0b01011010
        0b10000001

: target
        0b00111100
        0b01000010
        0b10011001
        0b10100101
        0b10100101
        0b10011001
        0b01000010
        0b00111100

#################################################
```

```
#  Register Map – Critical for RL extraction
################################################

:alias crosshair_x    v0  # Crosshair X position
:alias crosshair_y    v1  # Crosshair Y position
:alias score_reg      v2  # SCORE – RL agents read this!
:alias gameover_reg   v3  # GAME OVER FLAG (0=playing, 1=over)
:alias target_x       v4  # Target X position
:alias target_y       v5  # Target Y position
:alias target_active  v6  # Target active flag
:alias temp1          v7  # Temporary register
:alias temp2          v8  # Temporary register
:alias shot_active    v9  # Shot in progress flag
:alias targets_total  va  # Total targets appeared (max 10)
:alias key_reg        vb  # Key input register
:alias target_timer   vc  # Timer for current target
:alias missed_targets vd  # Count of missed targets

:const MAX_TARGETS 10     # Game ends after 10 targets total
:const TARGET_SIZE 8      # Target sprite size
:const CROSSHAIR_SIZE 8   # Crosshair sprite size
:const POINTS_PER_HIT 1   # Points awarded per target hit
:const TARGET_TIMEOUT 60  # Frames before target disappears (~3 sec at
   20fps)

################################################
#  Main Game Entry Point
################################################

: main
  # Initialize game state
  score_reg       := 0   # Score starts at 0
  gameover_reg    := 0   # Game is not over
  targets_total   := 0   # No targets appeared yet
  missed_targets  := 0   # No missed targets yet
  target_active   := 0   # No target active initially
  shot_active     := 0   # No shot in progress
  target_timer    := 0   # Timer at 0

  # Initial crosshair position (center)
  crosshair_x := 28
  crosshair_y := 12

  clear

  # Draw initial UI
  draw-crosshair

  # Main game loop
  loop
    # Check if game should end (10 total targets)
    if targets_total == MAX_TARGETS then jump game-over

    # Spawn new target if none active
    if target_active == 0 then spawn-target

    # Check target timeout
    if target_active == 1 then check-target-timeout

    # Handle player input
    handle-input

    # Check for hit if shot was fired
    if shot_active == 1 then check-hit
```

```
    # Small delay for playability
    temp1 := 1
    delay := temp1
    wait-delay

  again

##################################################
#  Target Timeout Check
##################################################

: check-target-timeout
  # Decrement timer
  target_timer += -1

  # Check if timer expired
  if target_timer != 0 then return

  # Target timed out - count as miss
  draw-target  # Erase target
  target_active := 0
  missed_targets += 1

  # Brief sound to indicate miss
  temp1 := 1
  buzzer := temp1
;

##################################################
#  Game Over Handler
##################################################

: game-over
  gameover_reg := 1    # Set game over flag for RL agent

  # Flash screen to indicate game over
  temp1 := 0
  loop
    clear
    temp2 := 5
    delay := temp2
    wait-delay

    draw-crosshair
    if target_active == 1 then draw-target
    temp2 := 5
    delay := temp2
    wait-delay

    temp1 += 1
    if temp1 != 3 then
  again

  # Infinite loop - game is over
  loop
    # RL agent should detect gameover_reg == 1
  again

##################################################
#  Input Handling
##################################################

: handle-input
  # Save current position
  temp1 := crosshair_x
```

```
  temp2 := crosshair_y

  # Movement controls (WASD) - use consistent key codes
  key_reg := 7  # A key (left)
  if key_reg key then temp1 += -2

  key_reg := 9  # D key (right)
  if key_reg key then temp1 += 2

  key_reg := 5  # W key (up)
  if key_reg key then temp2 += -2

  key_reg := 8  # S key (down)
  if key_reg key then temp2 += 2

  # Boundary checking
  if temp1 >= 254 then temp1 := 0   # Left boundary (wrapping check)
  if temp1 >= 56 then temp1 := 56   # Right boundary
  if temp2 >= 254 then temp2 := 0   # Top boundary (wrapping check)
  if temp2 >= 24 then temp2 := 24   # Bottom boundary

  # Check if position changed
  if temp1 != crosshair_x then jump update-crosshair
  if temp2 != crosshair_y then jump update-crosshair

  # Check for shoot (E key)
  key_reg := 6
  if key_reg key then shot_active := 1

  return

: update-crosshair
  # Erase old crosshair
  i := crosshair
  sprite crosshair_x crosshair_y CROSSHAIR_SIZE

  # Update position
  crosshair_x := temp1
  crosshair_y := temp2

  # Draw new crosshair
  sprite crosshair_x crosshair_y CROSSHAIR_SIZE

  # Check shoot after movement
  key_reg := 6
  if key_reg key then shot_active := 1
;

#################################################
#  Target Management
#################################################

: spawn-target
  # Generate random position for target
  target_x := random 0x37  # 0-55 range
  target_y := random 0x17  # 0-23 range

  # Ensure minimum distance from edges
  if target_x <= 2 then target_x := 3
  if target_y <= 2 then target_y := 3

  target_active := 1
  target_timer := TARGET_TIMEOUT  # Set timeout timer
  targets_total += 1              # Increment total targets count
  draw-target
```

```
;

: draw-target
  i := target
  sprite target_x target_y TARGET_SIZE
;

: draw-crosshair
  i := crosshair
  sprite crosshair_x crosshair_y CROSSHAIR_SIZE
;

##################################################
#  Hit Detection
##################################################

: check-hit
  shot_active := 0  # Reset shot flag

  # Check if target is active
  if target_active == 0 then return

  # Simple hit detection - check if crosshair center is near target
      center
  # Calculate X distance
  temp1 := crosshair_x
  temp1 += 4  # Crosshair center
  temp2 := target_x
  temp2 += 4  # Target center

  # Check X proximity
  if temp1 > temp2 then jump check-x-greater

  # crosshair is left of or at target
  temp2 -= temp1
  if temp2 > 6 then return  # Too far
  jump check-y-axis

: check-x-greater
  # crosshair is right of target
  temp1 -= temp2
  if temp1 > 6 then return  # Too far

: check-y-axis
  # Calculate Y distance
  temp1 := crosshair_y
  temp1 += 4  # Crosshair center
  temp2 := target_y
  temp2 += 4  # Target center

  # Check Y proximity
  if temp1 > temp2 then jump check-y-greater

  # crosshair is above or at target
  temp2 -= temp1
  if temp2 > 6 then return  # Too far
  jump register-hit

: check-y-greater
  # crosshair is below target
  temp1 -= temp2
  if temp1 > 6 then return  # Too far

: register-hit
  # Hit confirmed!
```

```
  # Erase target
  draw-target
  target_active := 0
  target_timer := 0  # Clear timer

  # Update score (for RL agent)
  score_reg += POINTS_PER_HIT

  # Sound feedback
  temp1 := 3
  buzzer := temp1
;

#################################################
#  Utility Functions
#################################################

: wait-delay
  loop
    temp1 := delay
    if temp1 != 0 then
  again
;
```

### F.2.3 LEVEL 3: MOVING TARGETS WITH TIME CONSTRAINTS

The most challenging level combines target movement with time limits, requiring predictive aiming and rapid response times.

Listing 4: Level 3 Target Shooter - Moving targets with time constraints

```
#################################################
#
#  Target Shooter Level 3 - RL Training Game
#
#  A deterministic shooting game designed for
#  reinforcement learning research.
#
#  Controls:
#  - WASD to move crosshair
#  - E to shoot
#
#  Score is stored in register v2 (score_reg)
#  Game over flag in register v3 (gameover_reg)
#
#  Features:
#  - Moving targets that bounce off walls
#  - Targets disappear after ~3 seconds if not hit
#  - Game ends after 10 targets (hit or missed)
#
#################################################

# Sprite data
: crosshair
        0b10000001
        0b01011010
        0b00100100
        0b01011010
        0b01011010
        0b00100100
        0b01011010
        0b10000001

: target
```

```
        0b00111100
        0b01000010
        0b10011001
        0b10100101
        0b10100101
        0b10011001
        0b01000010
        0b00111100

#################################################
#  Register Map - Critical for RL extraction
#################################################

:alias crosshair_x    v0  # Crosshair X position
:alias crosshair_y    v1  # Crosshair Y position
:alias score_reg      v2  # SCORE - RL agents read this!
:alias gameover_reg   v3  # GAME OVER FLAG (0=playing, 1=over)
:alias target_x       v4  # Target X position
:alias target_y       v5  # Target Y position
:alias target_active  v6  # Target active flag
:alias temp1          v7  # Temporary register
:alias temp2          v8  # Temporary register
:alias shot_active    v9  # Shot in progress flag
:alias targets_total  va  # Total targets appeared (max 10)
:alias key_reg        vb  # Key input register
:alias target_timer   vc  # Timer for current target
:alias target_vx      vd  # Target X velocity
:alias target_vy      ve  # Target Y velocity

:const MAX_TARGETS 10      # Game ends after 10 targets total
:const TARGET_SIZE 8       # Target sprite size
:const CROSSHAIR_SIZE 8    # Crosshair sprite size
:const POINTS_PER_HIT 1    # Points awarded per target hit
:const TARGET_TIMEOUT 80   # Frames before target disappears (~4 sec with
    movement)

#################################################
#  Main Game Entry Point
#################################################

: main
  # Initialize game state
  score_reg      := 0   # Score starts at 0
  gameover_reg   := 0   # Game is not over
  targets_total  := 0   # No targets appeared yet
  target_active  := 0   # No target active initially
  shot_active    := 0   # No shot in progress
  target_timer   := 0   # Timer at 0
  target_vx      := 0   # No initial velocity
  target_vy      := 0   # No initial velocity

  # Initial crosshair position (center)
  crosshair_x := 28
  crosshair_y := 12

  clear

  # Draw initial UI
  draw-crosshair

  # Main game loop
  loop
    # Check if game should end (10 total targets)
    if targets_total == MAX_TARGETS then jump game-over
```

```
    # Spawn new target if none active
    if target_active == 0 then spawn-target

    # Update target position if active
    if target_active == 1 then move-target

    # Check target timeout
    if target_active == 1 then check-target-timeout

    # Handle player input
    handle-input

    # Check for hit if shot was fired
    if shot_active == 1 then check-hit

    # Small delay for playability
    temp1 := 1
    delay := temp1
    wait-delay

  again

##################################################
#   Target Movement
##################################################

: move-target
  # Erase target at current position
  draw-target

  # Update X position
  target_x += target_vx

  # Check X boundaries and bounce
  if target_x >= 250 then jump bounce-left    # Hit left edge
  if target_x >= 56 then jump bounce-right     # Hit right edge

: check-y-movement
  # Update Y position
  target_y += target_vy

  # Check Y boundaries and bounce
  if target_y >= 250 then jump bounce-top      # Hit top edge
  if target_y >= 24 then jump bounce-bottom    # Hit bottom edge

: finish-move
  # Draw target at new position
  draw-target
  return

: bounce-left
  target_x := 1
  target_vx := 1  # Reverse to move right
  jump check-y-movement

: bounce-right
  target_x := 55
  target_vx := 255  # -1 to move left
  jump check-y-movement

: bounce-top
  target_y := 1
  target_vy := 1  # Reverse to move down
  jump finish-move
```

```
: bounce-bottom
  target_y := 23
  target_vy := 255  # -1 to move up
  jump finish-move

#################################################
#  Target Timeout Check
#################################################

: check-target-timeout
  # Decrement timer
  target_timer += -1

  # Check if timer expired
  if target_timer != 0 then return

  # Target timed out - count as miss
  draw-target  # Erase target
  target_active := 0

  # Brief sound to indicate miss
  temp1 := 1
  buzzer := temp1
;

#################################################
#  Game Over Handler
#################################################

: game-over
  gameover_reg := 1    # Set game over flag for RL agent

  # Flash screen to indicate game over
  temp1 := 0
  loop
    clear
    temp2 := 5
    delay := temp2
    wait-delay

    draw-crosshair
    if target_active == 1 then draw-target
    temp2 := 5
    delay := temp2
    wait-delay

    temp1 += 1
    if temp1 != 3 then
  again

  # Infinite loop - game is over
  loop
    # RL agent should detect gameover_reg == 1
  again

#################################################
#  Input Handling
#################################################

: handle-input
  # Save current position
  temp1 := crosshair_x
  temp2 := crosshair_y
```

```
  # Movement controls (WASD) - use consistent key codes
  key_reg := 7  # A key (left)
  if key_reg key then temp1 += -2

  key_reg := 9  # D key (right)
  if key_reg key then temp1 += 2

  key_reg := 5  # W key (up)
  if key_reg key then temp2 += -2

  key_reg := 8  # S key (down)
  if key_reg key then temp2 += 2

  # Boundary checking
  if temp1 >= 254 then temp1 := 0   # Left boundary (wrapping check)
  if temp1 >= 56 then temp1 := 56   # Right boundary
  if temp2 >= 254 then temp2 := 0   # Top boundary (wrapping check)
  if temp2 >= 24 then temp2 := 24   # Bottom boundary

  # Check if position changed
  if temp1 != crosshair_x then jump update-crosshair
  if temp2 != crosshair_y then jump update-crosshair

  # Check for shoot (E key)
  key_reg := 6
  if key_reg key then shot_active := 1

  return

: update-crosshair
  # Erase old crosshair
  i := crosshair
  sprite crosshair_x crosshair_y CROSSHAIR_SIZE

  # Update position
  crosshair_x := temp1
  crosshair_y := temp2

  # Draw new crosshair
  sprite crosshair_x crosshair_y CROSSHAIR_SIZE

  # Check shoot after movement
  key_reg := 6
  if key_reg key then shot_active := 1
;

##################################################
#  Target Management
##################################################

: spawn-target
  # Generate random position for target
  target_x := random 0x37  # 0-55 range
  target_y := random 0x17  # 0-23 range

  # Ensure minimum distance from edges
  if target_x <= 2 then target_x := 3
  if target_y <= 2 then target_y := 3

  # Generate random velocity (-1, 0, or 1 for each axis)
  target_vx := random 0x03
  if target_vx == 2 then target_vx := 255  # Convert 2 to -1

  target_vy := random 0x03
  if target_vy == 2 then target_vy := 255  # Convert 2 to -1
```

```
  # Ensure target is moving (not both velocities zero)
  if target_vx == 0 then jump ensure-movement
  jump finish-spawn

: ensure-movement
  if target_vy == 0 then target_vy := 1

: finish-spawn
  target_active := 1
  target_timer := TARGET_TIMEOUT  # Set timeout timer
  targets_total += 1              # Increment total targets count
  draw-target
;

: draw-target
  i := target
  sprite target_x target_y TARGET_SIZE
;

: draw-crosshair
  i := crosshair
  sprite crosshair_x crosshair_y CROSSHAIR_SIZE
;

#################################################
#  Hit Detection
#################################################

: check-hit
  shot_active := 0  # Reset shot flag

  # Check if target is active
  if target_active == 0 then return

  # Simple hit detection - check if crosshair center is near target
      center
  # Calculate X distance
  temp1 := crosshair_x
  temp1 += 4  # Crosshair center
  temp2 := target_x
  temp2 += 4  # Target center

  # Check X proximity
  if temp1 > temp2 then jump check-x-greater

  # crosshair is left of or at target
  temp2 -= temp1
  if temp2 > 6 then return  # Too far
  jump check-y-axis

: check-x-greater
  # crosshair is right of target
  temp1 -= temp2
  if temp1 > 6 then return  # Too far

: check-y-axis
  # Calculate Y distance
  temp1 := crosshair_y
  temp1 += 4  # Crosshair center
  temp2 := target_y
  temp2 += 4  # Target center

  # Check Y proximity
  if temp1 > temp2 then jump check-y-greater
```

```
  # crosshair is above or at target
  temp2 -= temp1
  if temp2 > 6 then return  # Too far
  jump register-hit

: check-y-greater
  # crosshair is below target
  temp1 -= temp2
  if temp1 > 6 then return  # Too far

: register-hit
  # Hit confirmed!
  # Erase target
  draw-target
  target_active := 0
  target_timer := 0  # Clear timer

  # Update score (for RL agent)
  score_reg += POINTS_PER_HIT

  # Sound feedback
  temp1 := 3
  buzzer := temp1
;

##################################################
#   Utility Functions
##################################################

: wait-delay
  loop
    temp1 := delay
    if temp1 != 0 then
  again
;
```

### F.2.4 ENVIRONMENT INTEGRATION AND WRAPPER IMPLEMENTATION

Once the LLM generates the CHIP-8 assembly code for each difficulty level, the games require integration with OCTAX's reinforcement learning interface. The environment wrapper extracts reward signals and termination conditions from the consistent register mapping established during code generation.

The Target Shooter implementation demonstrates the integration between LLM-generated content and the OCTAX framework. Each level maintains identical register assignments to ensure compatibility across the difficulty progression, enabling curriculum learning experiments without code modifications.

Listing 5: Target Shooter environment wrapper implementation

```python
from octax import EmulatorState

def score_fn(state: EmulatorState) -> float:
    """
    Extract score from register V[2]
    Score increments by 1 for each successful hit
    Range: 0-10 points
    """
    return state.V[2]

def terminated_fn(state: EmulatorState) -> bool:
    """
    Check game termination flag in register V[3]
```

```
    Game ends after 10 total targets (hit or missed in levels 2-3)
    """
    return state.V[3] == 1

# CHIP-8 key mapping for controls
# W=5 (up), A=7 (left), S=8 (down), D=9 (right), E=6 (shoot)
action_set = [5, 7, 8, 9, 6]

metadata = {
    "title": "Target Shooter - LLM-Generated RL Environment",
    "authors": ["Fully LLM-Generated Environment"],
    "description": "AI-generated progressive difficulty environment",
    "roms": {
        "target_shooter_level1": {
            "file": "target_shooter_level1.ch8",
            "description": "Static targets - Basic aiming skills"
        },
        "target_shooter_level2": {
            "file": "target_shooter_level2.ch8",
            "description": "Time-limited static targets"
        },
        "target_shooter_level3": {
            "file": "target_shooter_level3.ch8",
            "description": "Moving time-limited targets"
        }
    }
}
```

The consistent register mapping across all three levels enables direct comparison of agent performance and facilitates automated curriculum progression. Register V[2] consistently stores the score for reward calculation, while V[3] serves as the binary termination flag. The five-action control scheme (WASD movement plus shoot) provides sufficient complexity for interesting policies while remaining tractable for systematic analysis.

