# OpenReview forum: "Octax: Accelerated CHIP-8 Arcade Environments for Reinforcement Learning in JAX"
_ICLR.cc/2026/Conference — ICLR 2026 Poster_

### Official Review · Reviewer_E8PB · 2025-10-27

**Soundness:** 3
**Presentation:** 3
**Contribution:** 4
**Rating:** 8
**Confidence:** 5

**Summary:**

This paper introduces OCTAX, a suite of classic arcade-style RL environments implemented as a fully vectorized CHIP-8 emulator in JAX. The goal is to provide an Atari-like benchmark that (a) runs end-to-end on GPU, (b) scales to thousands of parallel environments, and (c) maintains authentic game dynamics. The authors show throughput up to ~350k steps/sec (≈1.4M frames/sec) with 8,192 parallel environments on a single consumer GPU (RTX 3090), which they report is roughly 14× faster than CPU-bound baselines like EnvPool on Atari Pong.

The benchmark covers 20+ CHIP-8 games across genres (puzzle, action, navigation, resource management, etc.), and the paper provides PPO training curves for 16 of them. These curves exhibit meaningful diversity in difficulty and learning dynamics (fast plateau vs. gradual improvement vs. failure-to-learn cases like Tetris/Worm), suggesting this is not just “16 clones of Breakout,” but a range of cognitive burdens.

Finally, the paper explores automatic environment generation: they use an LLM to synthesize new CHIP-8 games (e.g. a “Target Shooter” with increasing difficulty levels), plus score/termination logic, and then successfully train PPO on these generated tasks with clean difficulty gradients. This is pitched as a path toward scalable curriculum / rapid task creation.

Overall, the paper proposes OCTAX as “the missing GPU-native Atari for JAX RL,” arguing that this makes statistically reliable RL experiments cheaper and more reproducible for smaller labs.

**Strengths:**

**S1. Clear problem, clear value to the community.**
The paper correctly identifies a real bottleneck in RL: environments are still mostly CPU-bound, even though policy learning is GPU-bound. This prevents large-scale sweeps, high seed counts, and rigorous statistics, especially for vision-based tasks like Atari. OCTAX directly targets that gap with an image-based, arcade-style benchmark that runs entirely in JAX on GPU.

**S2. Strong engineering contribution.**
The authors implement a full CHIP-8 fetch/decode/execute loop in JAX using functional state passing, vectorized dispatch (lax.switch), and batched framebuffers, and wrap it as Gym/Gymnax-style RL envs with reward extraction, termination logic, action pruning, frame stacking, etc. They argue that fidelity to original game mechanics is preserved.
This is nontrivial and, as far as I know, not previously available in the JAX ecosystem for Atari-like games.

**S3. Throughput + scaling results are compelling.**
Reported numbers (350k steps/sec, linear-ish scaling up to 8k envs on a single 3090; 14× over EnvPool Pong at high parallelism) are impressive and very relevant to labs that don't have multi-node clusters. The memory footprint (2 MB per env, linear in env count) is also concretely discussed.
This is an unusually thorough systems evaluation for an RL benchmark paper.

**S4. Diversity of tasks and empirical characterization.**
They don’t just dump environments; they also measure PPO learning curves across 16 games and group them into qualitative regimes (fast plateau vs. gradual improvement vs. hard/sparse). This suggests these games could be used for algorithm diagnostics, curriculum, etc.

**Weaknesses:**

**W1. How “Atari-like” is it, really? (External validity.)**
CHIP-8 games are dramatically simpler than Atari 2600 in terms of resolution (64×32 monochrome), action semantics, and world complexity. The paper asserts “Atari-like cognitive demands,” but the qualitative gap (e.g., long-horizon exploration, partial observability, rich object interactions) is not deeply quantified. We see PPO struggling on Tetris/Worm, but I'd like more systematic evidence that success on OCTAX predicts anything on Atari, NetHack, Procgen, etc.
Right now OCTAX looks great for intra-JAX benchmarking and ablations, but it's less clear if it’s meaningful as a drop-in Atari replacement for algorithm claims.

**W2. Reward shaping / termination extraction feels artisanal.**
For each game, they manually or semi-automatically infer score registers, life counters, game-over flags, menu skips, etc. (e.g. “Brix stores score in V5, Pong encodes BCD in V14, etc.”). This is powerful, but also fragile and slightly underspecified.
If a new contributor adds a weird ROM, how confident are we that OCTAX's automatic heuristics won't silently produce a broken reward function (e.g. rewarding losing health)? The paper mentions static+dynamic analysis plus some LLM help, but I’d like stronger guarantees or validation.

**W3. No baselines beyond PPO.**
All learning curves are PPO only. There's no DQN-style baseline, no lightweight world-model baseline, no offline RL baseline, etc. PPO is reasonable (and popular), but I'd like to know if these environments produce meaningful rankings across algorithms, or whether they’re PPO-biased (e.g. continuous control style tuning, frame stacking assumptions, frame-skip assumptions).
Right now we mainly learn “PPO can learn some of them.”

**Questions:**

**Generalization / external validity.**
Do you have any evidence that OCTAX performance correlates with Atari performance, Procgen performance, or other visual RL benchmarks? Even a tiny pilot (e.g. rank-correlation of seed-averaged scores across algorithms) would strengthen the “Atari alternative” positioning.

**EnvPool/CuLE comparison.**
Can you report CuLE numbers (GPU Atari) on the same GPU you used for OCTAX, or at least discuss why that wasn’t feasible? That would make the “14× faster” claim feel less like apples vs oranges. Also, have you tried running OCTAX on CPU only to show the CPU→GPU delta cleanly?

**Automatic reward/termination inference.**
For a new arbitrary CHIP-8 ROM (unseen by you), how robust is score/termination extraction? Do you have quantitative success rates for the static+dynamic heuristics or the LLM-generated wrappers? For RL practitioners, “plug in a ROM and it just works” is a killer feature—please convince us it's realistic.

---

> ### Author Response · Authors · 2025-11-20
>
> Thank you for your thorough review and constructive feedback.
>
> > Generalization / external validity. Do you have any evidence that OCTAX performance correlates with Atari performance, Procgen performance, or other visual RL benchmarks? Even a tiny pilot (e.g. rank-correlation of seed-averaged scores across algorithms) would strengthen the “Atari alternative” positioning.
>
> Thank you for this great suggestion. We are currently investigating potential correlations between Octax performance and standard visual RL benchmarks. At present, we do not have evidence of a systematic relationship with Atari or Procgen results. However, we note qualitative similarities: for example, the reward dynamics during learning of CleanRL’s PPO implementation on Atari Pong resemble those observed in our Pong environment. While this is not sufficient to claim external validity, it motivates deeper analysis. We will continue exploring rank correlation and other cross-benchmark comparisons.
>
>
> > EnvPool/CuLE comparison. Can you report CuLE numbers (GPU Atari) on the same GPU you used for OCTAX, or at least discuss why that wasn't feasible? That would make the "14× faster" claim feel less like apples vs oranges. Also, have you tried running OCTAX on CPU only to show the CPU→GPU delta cleanly?
>
> We tried running CuLE but faced the unable to compile the project unless downgrading Ubuntu to version 18 and Cuda to version 10.0. The provided Dockerfile points to an image removed from public registries, and installing an older version of nvcc and gcc is not sufficient to compile CuLE successfully.
>
> > Automatic reward/termination inference. For a new arbitrary CHIP-8 ROM (unseen by you), how robust is score/termination extraction? Do you have quantitative success rates for the static+dynamic heuristics or the LLM-generated wrappers? For RL practitioners, "plug in a ROM and it just works" is a killer feature—please convince us it's realistic.
>
> We added a small feasibility study (Appendix E) evaluating whether an LLM can recover reward and termination functions directly from CHIP-8 non-symbolic assembly without any information about the game. The model (gpt-4o-mini) reliably identified simple score registers similarly to our human-defined score but seems to struggle with multi-condition termination logic, indicating that LLMs can assist with basic reward extraction but still require human oversight for more complex cases. In a real-world setting, we expect that providing the LLM with additional guidance would significantly improve reliability, for example by integrating it with our ROM analyzer.

---

> > ### Comment · Reviewer_E8PB · 2025-11-24
> >
> > Thank you very much for your answer.
> > That solves almost all the concerns.
> > I will maintain my score.
> > I believe this is what all jax-RL people has been waiting.

---

> > > ### Author Response · Authors · 2025-11-24
> > >
> > > Thank you for your feedback and for maintaining your positive assessment. We're glad this work addresses a need in the JAX-RL community and look forward to supporting researchers with this tool.

---

### Official Review · Reviewer_VWtY · 2025-10-27

**Soundness:** 3
**Presentation:** 3
**Contribution:** 2
**Rating:** 2
**Confidence:** 4

**Summary:**

This paper presents Octax, a high-performance arcade-style benchmark for Reinforcement Learning in Jax. The paper utilizes end-to-end GPU training, allowing extremely high throughput. Primarily, this work looks to provide a viable alternative to the long-standing Atari benchmark, but using considerably less computational resources. Furthermore, the paper presents a way to quickly use LLMs to generate new environments, on top of the current set of games presented.

**Strengths:**

This paper aims to achieve the important goal of reducing the cost of running Reinforcement Learning experiments, clearly succeeding, achieving an impressive 1.4 million frames per second on consumer-grade hardware. The code already being open-source is also a nice positive. LLM-generated environments using the provided prompts provide a nice way for developers to test environments with specific properties if they wish.

**Weaknesses:**

While I appreciate the attempt to reduce the computational burden of the Atari, I feel that the authors missed many key reasons why the Atari benchmark became so popular, and many of the features that still make it so popular today. In its current form, I don’t believe this work is capable of replacing the Atari benchmark without major reforms for a few key reasons:

- **A clear, fixed way to evaluate** - One reason Atari is useful is that there is a relatively clear and fixed protocol for evaluation [1]. This means researchers know what environments they need to run, the settings they need to use, and know that their work will be comparable to others. Furthermore, there are precise guidelines on how scores should be reported [2]. The paper would benefit from being very clear about what environments are included in the benchmark, how many frames/timesteps algorithms should be trained for, and how results should be reported, with examples. Currently Figure 3 does not meet the same standards as many papers which use the Atari benchmark.
- **Aggregate Scores** - Atari is useful as a benchmark, as the performance of an algorithm can be boiled down to a single number or single graph. In Atari, this has become human-normalized IQM performance with 95\% confidence intervals, which provides a score for the given algorithm. This prevents games with different score magnitudes from dominating the final score. Currently, this appears to be missing, despite games having scores of different magnitudes.
- **No-op Starts, Sticky Actions and Random Actions** - In the Atari benchmark, features are provided that prevent the agent from exploiting determinism (preventing brute-force approaches). Specifically, no-op starts randomly uses up to 30 no-op steps at the start of episodes; sticky actions give a 25% chance for actions to be repeated, and agents are forced to take random actions 1% of the time. While Octax appears to have some environments that have randomness, it's unclear whether brute-force approaches could work in some environments.
- **JAX only** - While JAX has its advantages in speed, for a benchmark, I see it as a significant weakness if this benchmark is exclusive only to JAX users. A significant portion of RL researchers and users use other frameworks, such as PyTorch, which appear to be excluded from using this benchmark.
- **Historic data** - One major advantage of Atari is that a huge number of algorithms have been evaluated, making it useful to compare against new algorithms, while Octax only has PPO. Please consider adding more algorithms such as DQN [3], SAC [4], PQN [5], and also more state-of-the-art algorithms would be appreciated.
- **Game categorizations** - Currently, in Octax games are categorized into groups such as Puzzle, Action, etc. I think it would be more beneficial for the research community if environments were grouped by the aspects of the algorithm they challenge. In Atari, there are well known groups such as hard exploration (Montezuma’s Revenge), long term-credit assignment (Skiing) and many more. For a good example of this, please look at BSuite [6]. While Table 1 somewhat provides this, I still don’t think it is up to the standards of recent work. Furthermore, I’d appreciate a more detailed description of the tasks, including the frequency and magnitude of rewards. It is currently unclear if  Octax provides environments that are as challenging as Atari - for example, even after years of research, environments such as Pitfall and Montezuma’s Revenge are still extremely challenging.


[1] Machado, Marlos C., et al. "Revisiting the arcade learning environment: Evaluation protocols and open problems for general agents." Journal of Artificial Intelligence Research 61 (2018): 523-562.

[2] Agarwal, Rishabh, et al. "Deep reinforcement learning at the edge of the statistical precipice." Advances in neural information processing systems 34 (2021): 29304-29320.

[3] Mnih, Volodymyr, et al. "Playing atari with deep reinforcement learning." arXiv preprint arXiv:1312.5602 (2013).

[4] Haarnoja, Tuomas, et al. "Soft actor-critic algorithms and applications." arXiv preprint arXiv:1812.05905 (2018).

[5] Gallici, Matteo, et al. "Simplifying Deep Temporal Difference Learning." The Thirteenth International Conference on Learning Representations.

[6] Osband, Ian, et al. "Behaviour Suite for Reinforcement Learning." International Conference on Learning Representations.

**Questions:**

- Is this benchmark only usable to those who have algorithms written in JAX?
- Does this benchmark have a set of rigorous set of specifications (number of frames, number of games, way to compare different overall performance) to ensure that researchers can easily compare their work?
- How challenging are the existing environments in this benchmark? Can you benchmark some different algorithms and provide code for these in the repository?
- Do all environments have a source of stochasticity? Or can they be solved by brute-force style algorithms?

Also, it appears Line 79 has a mistake. While I'm strongly in favor of benchmarks which make research easier, I feel this work still has a long way to go before it could replace something like Atari, thus I cannot yet recommend acceptance.

---

> ### Author Response · Authors · 2025-11-20
>
> Thank you for your review and the concrete suggestions for improving our benchmark standards. Your points about evaluation protocols, aggregate scoring, and game categorization are well-taken. We clarify below that our primary contribution is a toolkit for transforming CHIP-8 ROMs into vectorized RL environments, rather than a fully standardized benchmark like Atari. However, we recognize the value of standardization and will incorporate some of your suggestions in the revised manuscript.
>
> Our framework enables researchers to load arbitrary CHIP-8 ROMs and convert them into GPU-accelerated environments without recoding games from scratch, as required, for example, in MinAtar introduced in Young & Tian (2019). By eliminating CPU-GPU data transfer overhead, the system enables larger-scale experimentation than traditional CPU-bound environments, thus allowing for more statistical rigor (via more reported seeds and reproducibility) and more systematic algorithmic studies (e.g., via comprehensive ablations). We slightly revised our introduction to remove any phrases that might suggest we are a complete alternative to Atari or a fully specified benchmark, and we also mention this limitation in the conclusion. The library is designed for extensibility, enabling users to easily modify reward functions, termination conditions, and other components for their specific research questions. We anticipate community contributions will expand the environment suite, making premature standardization difficult. Nevertheless, we will add specifications for our current games, including detailed environment cards (similar to the [Arcade Learning Environment documentation](https://ale.farama.org/environments/)) with reward ranges, task descriptions, available actions, and game categorization inspired by Behavior Suite categories (Osband et al., 2019), a work initiated in Appendix C.3.
>
> > Is this benchmark only usable to those who have algorithms written in JAX?
>
> While our benchmark is optimized for JAX-based algorithms to leverage full GPU parallelization, we provide a Gymnax-to-Gym wrapper that enables compatibility with any algorithm using the standard Gym interface. However, using this wrapper introduces CPU-GPU communication overhead that reduces throughput. Researchers can still benefit from our diverse game environments when using non-JAX algorithms, though they will not achieve the same level of computational efficiency.
>
> > Does this benchmark have a set of rigorous set of specifications?
>
> We are adding comprehensive game specifications in the revised manuscript, including reward ranges, task descriptions, and available actions for each game (inspired by the [Arcade Learning Environment documentation](https://ale.farama.org/environments/)). We will also categorize games inspired by Behavior Suite categories as you suggested. We do not aim to provide a fixed benchmark at this stage; however, we thank you for the suggestions that will improve clarity and comparability. This revision is already initiated in Appendix C.3.
>
> > Can you benchmark some different algorithms and provide code for these in the repository?
>
> Our PPO implementation code is already available in the repository (based on Rejax). We added PQN (Gallici et al., 2024) as an additional baseline to demonstrate the efficiency of value-based methods, and we show the result in Figure 3.
>
> > Do all environments have a source of stochasticity?
>
> CHIP-8 includes a random number generation instruction called RND. Our implementation generates random numbers using JAX's PRNG, ensuring controllable stochasticity: Environments are fully seedable through JAX's random key management, enabling reproducible experiments.
> Only games executing the RND instruction have a stochastic behavior. Games without RND calls are deterministic and could theoretically be solved via planning or brute-force search. However, large state spaces and long episode horizons make exhaustive search computationally prohibitive for most games. According to your suggestion, we now provide wrappers for no-op starts and sticky actions to prevent exploitation of determinism where needed.
>
> Thank you for reporting the typo, we will correct it in the revision.
>
> **References:**
> Gallici, M., Fellows, M., Ellis, B., Pou, B., Masmitja, I., Foerster, J. N., & Martin, M. (2024). Simplifying deep temporal difference learning. *arXiv preprint arXiv:2407.04811*.
>
> Faldor, M., Zhang, J., Cully, A., & Clune, J. (2024). Omni-epic: Open-endedness via models of human notions of interestingness with environments programmed in code. *arXiv preprint arXiv:2405.15568*.
>
> Young, K., & Tian, T. (2019). Minatar: An atari-inspired testbed for thorough and reproducible reinforcement learning experiments. *arXiv preprint arXiv:1903.03176*.
>
> Osband, I., Doron, Y., Hessel, M., Aslanides, J., Sezener, E., Saraiva, A., ... & Van Hasselt, H. (2019). Behaviour suite for reinforcement learning. *arXiv preprint arXiv:1908.03568*.

---

> > ### Comment · Reviewer_VWtY · 2025-11-20
> >
> > Thank you for answering my questions and implementing many of my suggestions, including stochasticity, benchmarking PQN, and the use for non-JAX algorithms.
> >
> > While your reply has improved my confidence in the paper, I still find the question of how much of a benchmark or not a point of contention. I think your reply helps, in that it clarifies that this isn't supposed to be a standardized benchmark, but rather a useful way for researchers to answer research questions by quickly running many experiments on custom environments.
> >
> > I do still find this a little troubling, though - while I acknowledge many researchers like to generate environments to answer specific questions, not testing across a fair, standardized benchmark still often leads to trouble. For example, even in Atari, some work has been tested on subsets of environments such as hard exploration tasks, and later been found to be heavily detrimental to tasks outside of that subset. I think fair, standardized benchmarks have been a large driver of progress in AI, and you are now saying that this isn't the goal of your paper.
> >
> > Given that many of my concerns were answered and the aforementioned reasons for why this paper still has utility, I will raise my score to a 4. I cannot raise my score higher and advocate for acceptance since this still lacks the key feature I believe to be highly important.

---

> > > ### Author Response · Authors · 2025-11-24
> > >
> > > Thank you for acknowledging the improvements and raising your score. We keep in mind your concern about standardized benchmarks and their importance to the field, for future development. We hope the community will find this contribution useful for accelerating their development process.

---

### Official Review · Reviewer_iK9J · 2025-11-01

**Soundness:** 3
**Presentation:** 3
**Contribution:** 3
**Rating:** 6
**Confidence:** 4

**Summary:**

The paper presents OCTAX, a JAX-native, vectorized CHIP-8 emulator and RL environment suite that runs many GPU-accelerated game instances. OCTAX exposes (21) CHIP-8 titles as Gym/Gymnax-compatible environments, includes wrappers to extract score and termination signals, reports high throughput (claims up to hundreds of thousands env steps/s / millions of frames/s), provides PPO training experiments across multiple games, and demonstrates an LLM-assisted pipeline to generate CHIP-8 games and corresponding reward/termination wrappers. An anonymized code repository is provided.

**Strengths:**

The paper addresses a clear practical gap in the JAX ecosystem: image-based, GPU-native environments for RL. It thereby comes with significant engineering effort: a fully vectorized CHIP-8 emulator in JAX that can execute many parallel instances and integrates with standard RL training loops (Gym/Gymnax). Furthermore, this work directly empowers the community to work on more complex JAX-accelerated environments, which wasn't directly possible beyond Craftax before.

It introduces a broad catalog of 21 CHIP-8 games across multiple genres, useful for rapid prototyping and curriculum experiments. The reported throughput and scaling is promising, and if validated could substantially reduce wall-clock time for many experiments allowing for low resource algorithmic developments.

Finally, the authors for provide evidence for a novel auxiliary idea: an LLM-assisted pipeline to generate lightweight CHIP-8 games and wrappers, enabling rapid environment prototyping. I find this especially exciting since it may enable open-ended training and novel autocurricula approaches.

**Weaknesses:**

Emulation fidelity claims ('perfect fidelity') may be partially unsubstantiated: no instruction-trace equivalence, frame-by-frame comparison, or unit tests against a trusted CHIP-8 interpreter are reported. Most of the functional correctness assertions come from the successful training of agents.

Arguably, there is no really fair CPU baseline for CHIP-8 measured on the same machine, and comparisons to EnvPool/ALE conflate environment complexity differences. I understand that this is not trivial to accomplish, but I think it would make sense to maybe show runtimes of CPU/GPU-enabled environments just for contextualization. Even if they are not the same as the ones implemented in OCTAX.

The GPU memory/accounting claims (~2 MB per environment) lack a principled breakdown and appear implausible without explanation of XLA/JAX buffer and compiled executable overheads.

The LLM-assisted game generation is presented as a single case study with no statistics on compile/run success rates, human edit frequency, or failure modes.

Some broader claims (energy savings, enabling small labs) are asserted without corresponding measured energy or cost data. This could and should be better substantiated.

All experiments only consider PPO as the single RL algorithm tested.

**Questions:**

Re Emulator fidelity: Do you have systematic validation tests? E.g. for something like Pong, can you provide instruction-by-instruction equivalence tests, frame-by-frame rendering comparisons, and unit tests across representative ROMs against a trusted CHIP-8 interpreter.

Baselines and fairness: If you retain EnvPool/ALE comparisons, justify differences in environment complexity or compare EnvPool on a matched low-resolution workload. Additionally, consider adding DQN-style results.

Profiling and memory breakdown: Provide GPU profiler traces (kernel times, GPU utilization) and a detailed memory breakdown per environment (state arrays, framebuffers, intermediate buffers, compiled executable memory). Explain how you measured ~2 MB/env and the causes of scaling limits.

PPO protocol: Were hyperparameters tuned per game or held fixed? Provide ablations showing sensitivity to frame-skip and observation stacking and report whether reported learning curves use per-game tuning.

LLM pipeline evaluation: What fraction of LLM-generated ROMs compiled and ran without manual edits? How often did generated score_fn/terminated_fn require human correction? Provide statistics and typical failure cases and describe any automated validation used.

---

> ### Author Response · Authors · 2025-11-20
>
> Thank you for the detailed feedback and concrete suggestions for strengthening our contribution. We provide additional validation details and experimental results below to address these points.
>
> > Re Emulator fidelity: Do you have systematic validation tests?
>
> Unit tests are available in the repository for almost all instructions, and we test our emulator with a widely-used CHIP-8 test suite consisting of specialized ROMs designed to validate emulator fidelity: https://github.com/Timendus/chip8-test-suite. These tests are standard practice within the CHIP-8 community and provide comprehensive coverage of the instruction set.
>
> > Baselines and fairness
>
> We tried to add MinAtar+EnvPool for the low-resolution workload; unfortunately, MinAtar is not supported natively in EnvPool. To make it work, a dedicated wrapper would need to be implemented using EnvPool interfaces in C++. Among all the environments natively supported by EnvPool, Atari is the closest to us: some of our CHIP-8 games are functionally similar to their Atari counterparts (like Pong, Brix, Blinky, Tetris,...), and both provide discrete action spaces and frame-based observations, although our environments have reduced resolution and a monochrome display. We added PQN (Gallici et al., 2024) as an additional baseline in Figure 3 to complement our PPO results.
>
> > Profiling and memory breakdown
>
> We added the NVIDIA Nsight Systems profiler output to our anonymous repository (https://anonymous.4open.science/r/octax-C8E8/ablations/memory_profiling_3090/). Note that all environments use the same data structure, and all CHIP-8 operations are executed at each step due to vectorization, which is why there is no performance difference between environments, both in execution speed and memory usage. We therefore provide our analysis only on Pong.
>
> We measured memory using nvidia-smi before and after environment initialization, dividing the delta by the number of environments. Memory usage scales linearly with the number of parallel environments in our benchmark script. The scaling bottleneck on our NVIDIA RTX 3090 (24GB VRAM) arises from two factors: compute saturation at high parallelization levels and memory capacity. Indeed, at 2^14 environments × 2 MB/env ≈ 32 GB, we exceed the available 24 GB VRAM. This memory constraint limits scaling to 8,192 environments on this hardware.
>
> > PPO protocol: Were hyperparameters tuned per game or held fixed? Provide ablations showing sensitivity to frame-skip and observation stacking, and report whether reported learning curves use per-game tuning.
>
> Our hyperparameter search is detailed in Appendix D. We ran PPO on Pong and applied the same hyperparameters across all other games without per-game tuning. We used standard frame-skip and observation stacking values from Atari environments, acknowledging that optimal values may be game-dependent.
>
> > LLM pipeline evaluation: What fraction of LLM-generated ROMs compiled and ran without manual edits? How often did the generated score_fn/terminated_fn require human correction? Provide statistics and typical failure cases and describe any automated validation used.
>
> We provided one LLM-generated ROM, Target Shooter, as a proof of concept. With the appropriate prompt, including tutorials and examples (see Appendix E.1), Claude Opus 4.1 generated Target Shooter successfully in a zero-shot manner. Initial attempts without this context were unsuccessful. We expect success rates to depend on game complexity and prompt specificity. While systematizing this process, as done in OMNI-EPIC (Faldor et al., 2024), represents an interesting research direction, our goal here was to demonstrate the feasibility and potential of this approach for future work. We added a small feasibility study (Appendix E) evaluating whether an LLM can recover reward and termination functions directly from CHIP-8 non-symbolic assembly. The model reliably identified simple score registers but struggled with multi-condition termination logic, indicating that LLMs can assist with basic reward extraction but still require human oversight for more complex cases.
>
>
> **References:**
> Gallici, M., Fellows, M., Ellis, B., Pou, B., Masmitja, I., Foerster, J. N., & Martin, M. (2024). Simplifying deep temporal difference learning. *arXiv preprint arXiv:2407.04811*.
>
> Faldor, M., Zhang, J., Cully, A., & Clune, J. (2024). Omni-epic: Open-endedness via models of human notions of interestingness with environments programmed in code. *arXiv preprint arXiv:2405.15568*.
>
> Young, K., & Tian, T. (2019). Minatar: An Atari-inspired testbed for thorough and reproducible reinforcement learning experiments. *arXiv preprint arXiv:1903.03176*.

---

> > ### Comment · Reviewer_iK9J · 2025-11-21
> > **Response to Rebuttal Comment**
> >
> > Dear Authors,
> > Thank you very much for answering my questions and providing additional evidence. I will raise my score from 6 to 8 and would be happy to see this paper and its open-source contribution published at ICLR 2025.

---

> > > ### Author Response · Authors · 2025-11-24
> > >
> > > Thank you very much for your positive feedback and for raising your score. We greatly appreciate your support and are happy to contribute this open-source tool to the community.

---

### Official Review · Reviewer_6XnA · 2025-11-03

**Soundness:** 3
**Presentation:** 3
**Contribution:** 2
**Rating:** 6
**Confidence:** 3

**Summary:**

In this paper, the authors present a JAX implementation, known as OCTAX, of the CHIP-8 platform amenable to GPU acceleration. Octax converts CHIP-8 ROMs into RL environments that can take advantage of vectorized JAX functions. Octax provides a wide selection of games of interest to RL researchers spanning a number of different genres and difficulty levels. The performance of Octax is shown to be substantially higher than traditional methods for environment simulation using parallel instances of games with EnvPool. The authors note that some CHIP-8 ROMs had difficult reward functions to implement and understand, but this process was made easier with the help of LLMs to study the machine code. By reversing this process, they were able to train LLMs to produce new games that could be added to the RL training environment.

**Strengths:**

- The performance improvement of Octax over vectorized environments is impressive and will decrease the training time for many researchers using JAX to engage in RL experiments. With a shorter training time on simpler games, many new experiments to understand the training dynamics and tradeoffs between the large number of RL hyperparameters may be investigated more effectively.
- On top of improved acceleration, the authors demonstrate the ability of LLMs to aid in the understanding of existing ROMs and the generation of new environments based on properly constructed prompts. I believe this may be a simplified and interesting sandbox to explore LLM coding in a way that is faster than traditional code generation procedures.
- Providing this code as a testbed will facilitate the construction of more complex environments, such as Super-CHIP8, to add further complexity and interest.

**Weaknesses:**

- The utility of arcade environments for SOTA RL has passed, so the real value is to use the environment to accelerate understanding of RL training dynamics and other metrics. Although the added challenge of generating new environments may mitigate this point.
- Supporting yet another training environment may have a limited impact and lower the contributions.
- There's no discussion regarding the success rate of generating ROMs using LLMs. I assume this is for the sake of space and to focus the discussion of the paper on the acceleration of arcade environments.

**Questions:**

- How many attempts were required to generate Target Shooter using Claude? Was the success rate?

---

> ### Author Response · Authors · 2025-11-20
>
> Thank you for your review. You raise an important point about the practical utility of arcade environments. While we agree that arcade games alone may not push SOTA RL algorithms, our goal is to provide a fast, accessible platform for investigating training dynamics and algorithmic properties, which are areas where rapid iteration is essential but often limited by computational costs. We will emphasize in the revised manuscript that our contribution addresses both the need for GPU-native environments in JAX and the broader goal of enabling more investigations of RL training dynamics.
>
> Additionally, recent work demonstrates the benefits of highly parallelized environments for training efficiency. The JAX ecosystem remains limited in image-based environments. For example, Gallici et al. (2024) ran their entire experimental setup on a single GPU with full parallelization, except for Atari experiments, where they reported only 3 seeds (standard practice for Atari due to computational cost) compared to 10 seeds for other environments. Our contribution aims to address this situation by providing massively parallel, fully GPU-native image-based environments that reduce the computational barriers for future RL research.
>
> Beyond new benchmarks, the LLM generation capability opens possibilities for accelerating research in curriculum learning, continual learning, and meta-learning that extend beyond traditional arcade benchmarks.
>
> > How many attempts were required to generate Target Shooter using Claude? Was the success rate?
>
> With the appropriate prompt, including tutorials and examples (see Appendix E.1), Claude Opus 4.1 generated Target Shooter successfully in a zero-shot manner. Initial attempts without this context and with a less powerful LLM were unsuccessful. We expect success rates to depend on game complexity and prompt specificity. While systematizing this process, as done in OMNI-EPIC (Faldor et al., 2024), represents an interesting research direction, our goal here was to demonstrate the feasibility and potential of this approach as a proof of concept for future work.
>
> **References:**
> Gallici, M., Fellows, M., Ellis, B., Pou, B., Masmitja, I., Foerster, J. N., & Martin, M. (2024). Simplifying deep temporal difference learning. *arXiv preprint arXiv:2407.04811*.
>
> Faldor, M., Zhang, J., Cully, A., & Clune, J. (2024). Omni-epic: Open-endedness via models of human notions of interestingness with environments programmed in code. *arXiv preprint arXiv:2405.15568*.

---

### Author Response · Authors · 2025-11-20

We thank all reviewers for their thoughtful and constructive feedback, which has strengthened our manuscript. In response to your comments, we have made the following key revisions:

- Added GPU profiler traces (NVIDIA Nsight Systems) and detailed memory analysis to the repository
- Included PQN as an additional baseline in Figure 3 to complement our PPO results
- Added a feasibility study (Appendix E) systematically evaluating LLM-based reward and termination extraction from CHIP-8 assembly
- Added detailed environment specifications for all games, including reward ranges, action spaces, and task descriptions
- Provided wrappers for no-op starts and sticky actions
- Revised the introduction to better position Octax as an extensible toolkit for creating GPU-accelerated RL environments from CHIP-8 ROMs, rather than a fully standardized benchmark

We believe these revisions address the core concerns raised.

---

### Meta-Review · Area_Chair_Worh · 2026-01-07

**Summary:**

Forthcoming in this paper, OCTAX is a JAX-based framework that implements a fully vectorized CHIP-8 emulator with arcade environments accelerating on GPUs. This represents some of reinforcement learning’s latest work. The main problem identified by reviewers came at five central points in its design: (1) It does not offer unified standardization versus established benchmarks like ALE, and thus needs to standardize evaluation methods without pre-fixed fixed standards (Reviewer VWtY); (2) The generalizability of results in terms of external validity (Reviewers VWtY, E8PB) to wider areas such as Atari or Procgen; (3) The limitation of existing algorithmic baselines beyond the PPO, (Reviewers iK9J, VWtY, E8PB); (4) Inadequate validation of the LLM-supported environment generation pipeline (Reviewers 6XnA, iK9J); (5) The framework's narrow structure constrains its usefulness to JAX users, making it difficult for future work to fit directly back to existing practice areas (Reviewer VWtY). The choice to recommend acceptance as a poster is made based on the general agreement that the engineering contribution is significant addressing a real computational bottleneck, albeit with continuing concerns with respect to standardization and external validity.

**Reviewer Concerns:**

**Rebuttal Addressed Concerns:**

- *Emulation fidelity* (iK9J): Authors submitted unit tests and validation to standard CHIP-8 test suite, which was acceptable to reviewers.
- *GPU profiling and memory decomposition* (iK9J): Authors contributed NVIDIA Nsight Systems profiler traces and detailed memory analysis to this repository.
- *Limited algorithmic baselines* (iK9J, VWtY, E8PB): Authors placed PQN as the additional baseline in Figure 3.
- *Stochasticity and exploitation of determinism* (VWtY): Authors provided wrappers in case of no-op starts and sticky actions.
- *LLM pipeline evaluation* (6XnA, iK9J): Authors included a feasibility study (Appendix E) that evaluated LLM-based reward and termination extraction, and clarified that Target Shooter has been generated with zero-shot extraction after adequate prompting.
- *Positioning as benchmark vs. toolkit* (VWtY): Authors revised the introduction to clarify that OCTAX is an extensible toolkit and not a completely standardized benchmark.

**Outstanding Concerns:**

- *Lack of standardization* (VWtY): Although authors added environment specifications and categorizations, OCTAX still does not have fixed evaluation protocols or cumulative grading methods (e.g., human-normalized IQM) plus historical baselines similar to ALE which are comparable to them. In its review of the work VWtY specifically mentioned that this remains a primary limitation.
- *External validity* (VWtY, E8PB): There was no evidence to suggest that OCTAX performance aligns with Atari, Procgen, or other visual RL standards. This drawback was recognized by authors.
- *Framework specificity* (VWtY): The main performance advantages are exclusive to JAX, which limits the applicability of the results to PyTorch-based researchers.
- *LLM robustness for arbitrary ROMs* (E8PB): The feasibility study indicated LLMs had trouble with complex termination logic, reinforcing the need of human oversight.

**Reviewer Scores:**

- **Reviewer 6XnA**: Score of 6 initially. No indication of a change in score following rebuttal is made. Would probably hold 6 because their questions about utility and LLM success rates were treated but not central as part of their evaluation.
- **Reviewer iK9J**: 6 (original score) → **8** (rebuttal score). The reviewer clearly indicated satisfaction with the further evidence and validation.
- **Reviewer VWtY**: Initial score 2 → **boosted to 4** after counterargument. The reviewer recognized enhancements but added that consistency was still lacking and thus they could not recommend acceptance.
- **Reviewer E8PB**: Initial score 8 → **remained 8**. The reviewer, on the other hand, found significant help in addressing their concerns and strongly agreed acceptance was a best.

---

### Decision · Program_Chairs · 2026-01-26

Accept (Poster)